

# Nitrogen-containing Secondary Organic Aerosols Formation by Acrolein Reaction with Ammonia/Ammonium

Zhijian Li[1], Sergey A. Nizkorodov[2], Hong Chen[1], Xiaohui Lu[1*], Xin Yang[1,3*], Jianmin Chen[1]

[1] Shanghai Key Laboratory of Atmospheric Particle Pollution and Prevention, Department of Environmental Science and Engineering, Fudan University, Shanghai 200433, China

[2] Department of Chemistry, University of California, Irvine, California 92697, USA

[3] Shanghai Institute of Pollution Control and Ecological Security, Shanghai 200092, China

*Correspondence to*: Xiaohui Lu (luxiaohui@fudan.edu.cn); Xin Yang (yangxin@fudan.edu.cn)

**Abstract.**

Ammonia-driven carbonyl-to-imine conversion is an important formation pathway to the nitrogen-containing organic compounds (NOC) in secondary organic aerosols (SOA). Previous studies have mainly focused on the dicarbonyl compounds as the precursors of light-absorbing NOC. In this work, we investigated whether acrolein could also act as a NOC precursor. Acrolein is the simplest α,β-unsaturated mono-carbonyl compound, and it is ubiquitous in the atmosphere. Experiments probing gas-phase and surface reactions of acrolein as well as bulk liquid-phase experiments were carried out to study the reactivity of acrolein towards ammonia and ammonium ions. Molecular characterization of the products based on gas chromatography mass spectrometry, high resolution mass spectrometry, surface enhanced Raman spectrometry and Ultraviolet/visible spectrometry was used to propose possible reaction mechanisms.

We observed 3-methyl pyridine (also called 3-picoline) in the gas phase in the Tedlar bag filled with gaseous acrolein and ammonia. In the liquid phase, oligomeric compounds with formulas $(C_3H_4O)_m(C_3H_5N)_n$ and pyridinium compounds like $(C_3H_4O)_2C_6H_8N^+$ were observed as the products of acrolein reaction with ammonium ions. The 3-picoline could be the product of acrolein reaction with gaseous ammonia in the gas phase. The pathway to 3-picoline was proposed to be the intramolecular carbon-carbon addition of the hemiaminal which resulted from sequential carbonyl-to-imine conversions of acrolein molecules. The similar reaction of dissolved acrolein with ammonium ions leads to the formation of 3-methyl pyridinium (also called 3-picolinium) cations in the liquid phase. The $(C_3H_4O)_2C_6H_8N^+$ was a carbonyl-to-hemiaminal product from acrolein dimer and 3-picolinium cations, while the oligomeric products of $(C_3H_4O)_m(C_3H_5N)_n$ were polymers



of acroleins and propylene imines via carbonyl-to-imine conversion and condensation reactions. Part of the 3-picolinium could re-volatilize to the gas phase as 3-picoline, explaining the observation of gaseous 3-picoline in the bag filled with acrolein and ammonium sulfates (or chlorides) aerosols. The pH value effect on the liquid products was also studied in the bulk liquid-phase experiments. Compared to the oligomeric compounds forming in both acid and alkaline conditions, there is

a tendency for the pyridinium products to be formed under moderately acidic conditions. Both the oligomeric products and the pyridinium salts are light absorbing materials. This work suggests a potential role for acrolein reaction with ammonia/ammonium as a source of light-absorbing heterocyclic NOC in SOA. Therefore, secondary reactions of α,β-unsaturated aldehydes with reduced nitrogen should be taken into account when evaluating climate and health effects of SOA.

## 1. Introduction

Carbonyl compounds are ubiquitous in the gaseous, liquid and particulate phases in the atmosphere (Carlier et al., 1986; Dai et al., 2012; Grosjean, 1982). The high reactivity of carbonyls makes them important intermediates in chemical conversion of volatile organic compounds (VOCs) into secondary organic aerosol (SOA) by multi-phase chemical processes.

After the uptake to the particulate phase, volatile carbonyls may undergo photooxidation (Renard et al., 2015), aldol condensation (Sareen et al., 2010; Yasmeen et al., 2010), oligomerization (De Haan et al., 2011; Renard et al., 2016; Shen et al., 2016), imine (R=NH compound) or hemiaminal (R(OH)-NR$_2$ compound) formation (Maxut et al., 2015; Yu et al., 2011) or Mannich reaction (Noziere and Cordova, 2008; Wang et al., 2010) to form high-molecular-weight (high-MW) compounds in SOA. For example, compounds that possess two carbonyl groups (dicarbonyls), such as glyoxal and methylglyoxal, are

important precursors of aqueous SOA (Galloway et al., 2014; Shapiro et al., 2009; Trainic et al., 2011; Volkamer et al., 2007; Yu et al., 2011; Zhao et al., 2006). A number of laboratory and field studies have highlighted the importance of carbonyl-to-imine reaction for the formation of light absorbing nitrogen-containing organic compounds (NOC) from dicarbonyls (De Haan et al., 2011; Hawkins et al., 2018; Lin et al., 2015). Since the two carbonyl groups in dicarbonyls can both undergo the carbonyl-to-imine formation, heterocyclic aromatic NOC derived from pyrrole, imidazole, pyridine, etc. are

formed (Aiona et al., 2017; Bones et al., 2010; Flores et al., 2014; Kampf et al., 2012; Kampf et al., 2016; Laskin et al., 2010;





O'Brien et al., 2013; Teich et al., 2016; Updyke et al., 2012). The extended conjugation in the resulting NOC compounds leads to the light-absorptivity of the SOA particles containing these compounds (Lee et al., 2013).

Mono-carbonyl compounds such as formaldehyde, acetaldehyde, hydroxyacetone and acrolein are important precursors of free radicals, peroxyacetyl nitrates (PAN), ozone and other reactive oxygen species (Mellouki et al., 2015). They can undergo the acid-catalyzed heterogeneous reactions after being taken up into the acidic aerosols (Garland et al., 2006; Jang et al., 2002). Mono-carbonyls can also be converted into light-absorbing SOA compounds via multi-phase chemistry. For example, Powelson et al. (2014) investigated the formation of light-absorbing SOA in aqueous reactions of different mono-carbonyl compounds including formaldehyde, acetaldehyde, hydroxyacetone and several dicarbonyl compounds with amines or ammonium salts. Most carbonyl compounds are able to undergo aldol condensation under highly-acidic conditions to form oligomers which absorb light in the 300-500 nm range (Noziere and Esteve, 2007; Van Wyngarden et al., 2015). The previous experiments suggest that the light absorptivity of SOA derived from mono-carbonyl compounds is in general smaller than that from reactions of glyoxal and methyl glyoxal. However, mono-carbonyl compounds such as formaldehyde and acetaldehyde were found to increase the production of imidazoles by dicarbonyls (Rodriguez et al., 2017).

The majority of the reactions of carbonyl compounds leading to light-absorbing products involved saturated aldehydes and ketones. Unsaturated mono-carbonyl compounds have received less attention in this regard. Nozière et al. (2006) have investigated the uptake of unsaturated mono-carbonyl compounds such as methyl vinyl ketone and methacrolein onto sulfuric acid solution via aldol condensation. In α,β-unsaturated mono-carbonyl compounds, such as acrolein, the conjugation of the carbonyl group with the alkene, results in a charge distribution within the molecule that makes both nucleophilic addition and electrophilic addition possible. The reactivity of α,β-unsaturated mono-carbonyl compounds with amines and ammonia have not been previously explored in an atmospheric chemistry context.

Acrolein is a typical α,β-unsaturated mono-carbonyl compound with a widespread occurrence in the atmosphere. Acrolein was observed at concentrations comparable to those of acetaldehyde in a coastal region of Southern Europe (Cerqueira et al., 2003; Romagnoli et al., 2016). According to ambient measurements by Altemose et al. (2015), the average concentration of acrolein was $2.9 \pm 0.8$ μg m$^{-3}$ during Beijing Olympics. Cahill (2014) found that even the natural background concentration of acrolein in summer in California was higher than 20 ng m$^{-3}$. Acrolein has numerous sources, including vehicular emissions

(Grosjean et al., 2001), wood combustion (Lipari et al., 1984), cooking oil heating (Umano and Shibamoto, 1987), smoking (Feng et al., 2006) and other sources (Ho and Yu, 2002; Reda et al., 2015; Seaman et al., 2007). It is also one of the products of atmospheric photooxidation of other VOCs such as 1,3-butadiene (Baker et al., 2005; Liu et al., 1999; Tuazon et al., 1999). The OH-initiated oxidation of acrolein in the presence of $NO_x$ may also lead to the formation of PAN-type species

(Magneron et al., 2002; Orlando and Tyndall, 2002). Reactive uptake of acrolein on aerosols containing ammonium sulfate, if effective, could potentially act as an alternative sink for acrolein as well as a source of particulate organic compounds, including NOC.

The goal of this work is to improve our understanding of SOA formation from α,β-unsaturated carbonyls. We are reporting results of experiments on reactive uptake of acrolein by aqueous aerosols as well as bulk solutions containing ammonium

sulfate, the most abundant inorganic salt in the atmosphere. Possible mechanisms of chemical reactions of acrolein with ammonia in gas phase and with ammonia or ammonium ion in liquid phase are proposed based on the molecular characterization of the reaction products using mass spectrometry. The results suggest that acrolein + ammonia/ammonium reactions may lead to low-volatility NOC compounds in SOA.

## 2. Experiments

### 2.1 Tedlar bag experiments

Tedlar bag experiments were carried out to study the reaction of acrolein with ammonia/ammonium. Four inflatable Tedlar bags with maximum volumes of 100 L were prepared. One was labeled as acrolein-ammonia bag, two were labeled as acrolein-ammonium bags, and one was used for control experiments. We prepared a 0.15 M acrolein solution in water, a 2.2

M ammonium hydroxide solution, and 1 M $(NH_4)_2SO_4$ and 2 M $NH_4Cl$ aqueous solutions in advance. By evaporating 2 mL of the acrolein solution and 400 μL of the ammonium hydroxide solution in the water bath at 40°C, nearly 5 mmol ammonia and 0.3 mmol acrolein were introduced into the acrolein-ammonia bag. For the acrolein-ammonium bags, 2 mL of the acrolein solution was evaporated into each bag as well. However, particles containing ammonium ions were added to the acrolein-ammonium bag instead of gaseous ammonia. These particles were generated by aerosolizing about 2 mL of 1 M

$(NH_4)_2SO_4$ or 2 M $NH_4Cl$ aqueous solutions (about 4 mmol ammonium ions) in a TSI atomizer (Model 9302). The size of





the generated aerosols peaked around 800 nm, as shown in Figure S1. The size distribution was measured by a TSI Aerodynamic Particle Sizer Spectrometer (Model 3321). Dry $N_2$ from the gas cylinders was used for filling the bags to their maximum volume of 100 L. The relative humidity (RH) in the bags was measured as 90-100%. The bags were stored under dark conditions at room temperature for 2 hours prior to analysis.

The 4 mL of water evaporated/aerosolized in the 100 L bag volume in acrolein-ammonium experiments provided more than enough water to maintain saturated conditions with respect to water vapor and sustain a liquid film on the bag walls. The aerosolized $(NH_4)_2SO_4$ or $NH_4Cl$ particles must have quickly deposited to the bag walls forming a coating on the surface. Because the RH was higher than the deliquescence points (80%) of the salts (Takahama et al., 2007), the wall coating likely existed as an aqueous film containing dissolved ammonium ions on the inner surface of the bag. Therefore, both the

gas-phase constituents and the constituents of the aqueous film were collected for analysis, in order to investigate the gas-phase reaction and liquid-phase reaction, respectively. The RH in the acrolein-ammonia experiments was also high, around 90%. Because of the high RH, the acrolein and ammonia in the acrolein-ammonia bag could partition into both the gas phase and the liquid film on the bag wall (The Henry's constants for acrolein and for ammonia are $\sim 8$ mol $L^{-1}$ atm$^{-1}$ and $\sim 60$ mol $L^{-1}$ atm$^{-1}$ respectively.). Therefore, the wall residues were analyzed in these experiments as well.

The gas-phase constituents in the bags were analyzed by gas chromatography mass spectrometry (GC-MS, Thermo Focus DSQ). A solid-phase microextraction (SPME) fiber (50/30μm DVB/CAR/PDMS 2cm, Supelco) was directly inserted into the bag for 15 min. The SPME-collected constituents were desorbed at 250 °C for 2min, then measured by GC-MS. The temperature program used for GC-MS analysis was as follows: 60 °C for 3 min, followed by a linear ramp to 300 °C at 30 °C min$^{-1}$, followed by 2 min at 300 °C. Helium (99.999%) was the carrier gas maintained at a flow-rate of 1 mL min$^{-1}$. A

splitless mode was used. The electron impact energy was 70 eV and the mass-to-charge range was *m/z* 41-400 in the full-scan acquisition mode. Compounds were identified using the NIST Mass Spectral Library (National Institute of Standards and Technology, Washington, DC, USA). The basic properties of chemicals such as the Henry's constant of acrolein and picoline and their typical Ultraviolet/visible (UV/vis) spectra were obtained from NIST Chemistry WebBook, Standard Reference Data 69.

The constituents in the aqueous film on the bag walls were analyzed by electrospray ionization high resolution mass



spectrometer (ESI HRMS, Agilent 6540, with mass resolving power of 40,000). The residue remaining on the bag walls was rinsed by 1 mL acetonitrile/water (1/1, v/v) and then rapidly detected by direct infusion ESI HRMS in the positive mode. The parameters used for the mass spectrometer were the following: spray voltage 3000 V, sweep gas flow rate 0 respective arbitrary units (AU); sheath gas flow rate 10 AU; aux gas flow rate 5 AU; ion transfer tube temperature 350 °C; vaporizer

temperature 300 °C; scan range $m/z$ 50 − 500; maximum injection time 100 ms; automated gain control (AGC) target 250 000; S-lens RF level, 60%. The major ion peaks were assigned as protonated formulas $C_cH_{h+1}O_oN_n^+$ (c≥5, h≥5, 0≤o≤10, 0≤n≤10) using the Agilent MassHunter Qualitative Analysis B.07.00. The mass accuracy limit was set as ±3 ppm, and ratio of the double bond equivalent (DBE) value to the number of carbon atoms (C-number) was constrained to be less than 0.7. Since most compounds were detected as protonated ions, $C_cH_{h+1}O_oN_n^+$, DBE value of the neutral compound $C_cH_hO_oN_n$ was

calculated as c+1-(h-n)/2.

## 2.2 Bulk aqueous-phase experiments

Bulk aqueous-phase experiments were carried out to study the reaction of acrolein with ammonium ion in solution. Solutions containing 1 M of ammonium ion with different pH values were prepared by mixing different volumes of

ammonium hydroxide and hydrochloric acid. A volume of 25 μL of acrolein was then mixed into the 5 mL of the solution to achieve an initial acrolein concentration of 75 mM. All solution mixtures were kept in capped glass bottles in the dark at room temperature for 2 hours (For UV/vis study, the reaction time was set as 30 min, 1 hour, 2 hours and 4 hours).

Since the amount of sample was much larger than in the Tedlar bag experiments, several additional types of analysis including ultra-high-performance liquid chromatography (UPLC) coupled with ESI HRMS, surface enhanced Raman

spectrometry (SERS) and conventional UV/vis spectrophotometry could be applied to the constituents of the bulk sample at the end of reaction. The solutions after reaction were diluted by a factor of 20 with deionized water. The constituents were separated by an UPLC column (Agilent ZORBAX SB-C18 HD, 50 x 2.1 mm) and detected by ESI HRMS (Agilent 6540). The eluent composition was (A) Milli-Q grade water and (B) acetonitrile at a flow rate of 0.6 mL/min. The mobile phase gradient was initially 95:5 (v/v, A/B), increased to 100% B in 15 min and returned to 95:5 (v/v, A/B) in 1 min and then kept

for 4 min to re-equilibrate the column. We also did the formula assignment for the MS-measured reactants and products. For

SERS analysis, 500 μL of the solution was mixed with a certain concentration of gold nanoparticles, and 100 μL of the

mixture was then dropped on the silicon wafer and dried in air for measurement by a LabRam-1B Raman spectrometer.

Absorption spectra of the bulk samples were measured by a conventional UV/vis spectrophotometer (U-3000, Hitachi). The

scan range of the spectrometer was set to 190-700 nm, a quartz cuvette with an optical path length of 1 cm was used, and

deionized water was used as the reference solution.

## 3. Results and Discussion

### 3.1 Analysis of gaseous products of acrolein reaction with ammonia

Figure 1 shows GC traces for the gas-phase compounds in the acrolein-ammonia bag and in the control experiment (in

which only acrolein was introduced into the bag). For the control experiment, two species were separated at the retention

time 1.59 min and 4.27 min. With help of the NIST MS library, the one at 1.59 min was identified as acrolein, and the other

at 4.27 min as pyran aldehyde. Pyran aldehyde is a known acrolein dimer generated via Diels-Alder reaction (Reaction 1 in

Figure 2). The dimer is commonly found in acrolein-containing solutions and it is easier to quantify than acrolein (for

example, it was used as the marker for detection of acrolein in wine) (Bauer et al., 2010; Bauer et al., 2012). In the control

experiment, acrolein existed in the bag for 2 hours. The Diels-Alder reaction likely occurred on the wetted surface of the bag

or in the solution before injection into the bag, and the dimer evaporated back into the gas-phase (although we cannot rule

out gas phase reaction).

The chromatogram of the gaseous components in the acrolein-ammonia bag was different from that of the control

experiment. Pyran aldehyde was no longer observed. Two strong chromatographic peaks occurred at retention times of 1.60

min and 4.18 min. The former corresponds to acrolein while the latter could be identified as 3-methylpyridine, also known as

3-picoline. As a confirmation, 3-picoline standard was detected by the same GC-MS instrument at the same retention time,

shown in Figure 1(c).

In the organic chemical studies, it is known that pyridine and its derivatives can be synthesized from α,β-unsaturated

aldehydes or ketones with ammonia through multi-step chemistry, usually including imine formation, aldol condensation and

Michael reaction (Brody and Ruby, 1960; Stitz, 1942; Tschitschibabin, 1924; Tschitschibabin and Oparina, 1927; Zhang et



al., 2014; Zhang et al., 2016). However, all these reported syntheses in either gas-phase or liquid-phase required elevated temperatures and presence of an acidic solid-state catalyst. Formation of picoline from acrolein under room temperature conditions and without a catalyst has not been reported before. Kampf et al. (2016) have reported formation of pyridine as an organic aerosol compound, which has the same aromatic core as picoline, from glutaraldehyde (a 1,5-dicarbonyl compound)

undergoing sequential carbonyl-to-imine reactions, similar to the Paal-Knorr pyrrole formation from 1,4-dicarbonyls. However, this mechanism would not work for acrolein, which is a mono-carbonyl with only 3 carbon atoms. Dihydropyridine compounds were also found in the mixture of acetaldehyde, acetylacetone and ammonia sulfate by Kampf et al. (2016) Both carbonyl-to-imine conversion and condensation of acetaldehyde and acetylacetone derivatives contributed to the dihydropyridine formation.

The picoline formation in our work could also be a combined result of condensation and imine formation. We propose the pathway for the picoline formation in Reactions 2 in Figure 2. An acrolein molecule is converted into propylene imine by carbonyl-to-imine conversion and then undergoes the carbonyl-to-imine conversion again with another acrolein molecule to form a hemiaminal. The β-carbon in one alkenyl group has a lower electronegativity than the α-carbon in the other alkenyl group, leading to the intramolecular addition reaction between them. Thus, picoline is generated after the addition reaction

and the subsequent dehydration. This proposed pathway leads to the formation of 3-picoline, which is identified as the main product in our experiments, rather than its isomeric 2-picoline or 4-picoline. Moreover, the pathway is similar to the mechanism Stitz et al. (1942) proposed for the synthesis of 3-picoline in gas-phase, but Stitz's work involved the temperature higher than 110 °C and the catalyst. Our observation of similar reaction products under ambient temperature conditions suggests that conversion of acrolein into NOC may take pace without a catalyst under atmospherically relevant conditions.

Moreover, whether in the gas phase or in the liquid phase the proposed reaction occurs needs to be discussed. Since we did not detect any pyridinium compounds in the liquid film on the wall of the acrolein-ammonia bag (see section 3.2), the 3-picoline should be formed via direct gas-phase reaction of gaseous acrolein with gaseous ammonia. However, the GC-MS measurement results for the gaseous compounds in the acrolein-ammonium bag indicated that the reaction should occur in the liquid phase. As shown in Figure 1(d), the major gaseous components in the acrolein-ammonium bag were similar to

those in the control experiments. Acrolein and its dimer, pyran aldehyde, were detected at 1.61 min and 4.28 min as major

peaks. Picoline was also observed, but with a much smaller abundance than that in the acrolein-ammonia bag. Since ammonium sulfates/chlorides aerosols instead of gaseous ammonia were filled in the acrolein-ammonium bag, the gaseous 3-picoline detected in the acrolein-ammonium bag ought to result from liquid-to-gas partition equilibrium of the 3-methyl pyridinium compounds formed via Reaction 2 in the liquid phase. In fact, the pyridinium products were observed in the

liquid film on the wall of the acrolein-ammonium bag. (see section 3.2)

In addition to peaks for the reactant of acrolein and product of 3-picoline, four smaller peaks were observed at 1.50 min, 1.67 min, 2.89 min and 3.28 min in Figure 1(a). The MS pattern for the peak at 1.67 min failed to match any reasonable chemicals, but the remaining three were identified as acetaldehyde, pyridine and allyl acrylate, respectively. Some of acrolein likely underwent the aqueous-phase decomposition back to acetaldehyde and formaldehyde in the prepared solution

or in the aqueous film on the bag wall, leading to detection of acetaldehyde after liquid-to-gas phase partitioning. The allyl acrylate as a by-product could be attributed to the acrolein condensation, though the reported allyl acrylate formation by acrolein requires a catalyst (Youngman and Rust, 1961). The observed pyridine would require picoline demethylation, but how the demethylation happened is not clear yet.

### 3.2 Analysis of condensed-phase products of acrolein reaction with ammonium in the liquid film

Considering the high humidity in the bag and the liquid film formed on the bag wall, we examined the composition in the liquid film to study the condensed-phase products. The direct infusion ESI HRMS results for the bag-wall residue samples and for a standard acrolein solution are displayed in Figure 3. For acrolein itself, the spectra were dominated only by the

strong signal at $m/z$ 113.0597, corresponding to $[C_6H_9O_2]^+$ by the formula assignment. As pyran aldehyde $C_6H_8O_2$ is known as the main dimer of acrolein, and is also observed in the gaseous sample by GC-MS, the $[C_6H_9O_2]^+$ is assigned protonated pyran aldehyde.

The bag-wall residue samples had more peaks in their mass spectra, extending all the way to $m/z$ 500. We did the formula assignment for the top 30 peaks for each mass spectrum. The assignments are listed in Table S1 and shown in Figure 4 by

plotting the DBE value versus C-number in the assigned formulae of un-ionized species (the $H^+$ or $NH_4^+$ were removed from



the ion formula). The orderly distribution of the observed C-numbers in multiples of 3 in Figure 4 implies oligomerization of C3 compounds. The formula of acrolein is $C_3H_4O$, and propylene imine (the unstable intermediate product of reaction between acrolein and ammonia) has a formula $C_3H_5N$. The compounds in the residue samples appeared to represent the oligomerization products between several acrolein and propylene imine molecules, though the reaction pathway to the

oligomers was unclear. Simple addition oligomers with formulas $(C_3H_4O)_m(C_3H_5N)_n$ should have DBE of $m+n+1 = (1/3)\times$C-number + 1. Because the addition oligomers could be additionally hydrated or ammoniated, the observed DBE can deviate in the down direction from the $(1/3)\times$C-number+1 prediction. As shown in Figure 4, the majority of the assigned compounds were located in the region of "DBE$\leq$1/3 C-number + 1", suggesting that the $(C_3H_4O)_m(C_3H_5N)_n$ oligomers are the main products of liquid phase reaction between acrolein and ammonium. The ions detected in the bag-wall residues in

the acrolein-ammonia bag differed from those ions detected in the acrolein-ammonium bag (Figure 4). The bag-wall sample in the acrolein-ammonia bag contained the oligomers with lower molecular weight and comparatively more N atoms.

As we discussed in the last section, gaseous 3-picoline was abundant in the acrolein-ammonia bag, and it could potentially participate in the oligomerization reactions. Picoline-derived oligomers would have an even larger DBE than $(C_3H_4O)_m(C_3H_5N)_n$ oligomers of acrolein and propylene imine molecules. However, we did not observe any compounds with

DBE larger than "1/3$\times$ C-number + 1" for the bag-wall residue sample in the acrolein-ammonia bag. Even if picoline could partition into the liquid phase it did not appear to efficiently participate in oligomerization.

Nevertheless, the bag-wall residue sample in the acrolein-ammonium bags contained some organics located above the "DBE=1/3 C-number + 1" line in Figure 4, indicating the possible presence of some heterocyclic compounds. For example an ion at $m/z$ 206.1178, assigned to $[C_{12}H_{16}ON_2]^+$, had the highest abundance among all the compounds with DBE larger

than "1/3$\times$C-number + 1". Since picoline has the formula of $C_6H_7N$, the ions' formula could be $(C_3H_4O)_2C_6H_8N^+$, indicating the presence of pyridinium compounds in the liquid phase. We proposed that a reaction similar as Reaction 2 in Figure 2 leads to the formation of 3-methyl pyridinium cations, and then the 3-methyl pyridinium cations can attack the electrophilic site in the carbonyl of the acrolein dimer, to change the carbonyl to hemiaminal and form pyridinium compounds $(C_3H_4O)_2C_6H_8N^+$ (Reaction 3 in Figure 2).



### 3.3 pH values and liquid products of acrolein reaction with ammonium

As the analysis and discussion above demonstrated, organics including polymers $(C_3H_4O)_m(C_3H_5N)_n$ and pyridinium compounds were formed in the ammonium-containing liquid film while 3-picoline, as a gaseous pyridine derivative, were formed in the ammonia-rich gas phase. In the acrolein-ammonia bag, the liquid film on the bag-wall also contained

ammonium ions, but no pyridinium compounds were observed in the residue sample. The pH is the possible cause. On the one hand, the heterogeneous reactions of carbonyls with ammonia were reported as acid-catalyzed (Jang et al., 2003). On the other hand, as the pKa of 3-picoline is 5.68 (at 20 °C), the 3-picolinium compounds in the liquid phase tend to be neutral molecules of 3-picoline and then volatilize to the gas phase under a high pH condition. It also explained the observation of gaseous 3-picoline in the acrolein-ammonium bags in section 3.1. The $NH_4OH$ solution we injected into the bag was alkaline

while the $(NH_4)_2SO_4$ or $NH_4Cl$ solution had pH of ~ 6. In order to investigate the role of pH in the reaction of acrolein with ammonium, we carried out bulk aqueous experiments at different pH values.

The UPLC traces for the products observed in bulk aqueous experiments with various pH values are illustrated in Figure 5. The separated species were quite different for the highly acidic (pH of 2-4), moderate acidic and neutral (pH of 5-7), and alkaline (pH of 8-12).

At pH of 10 or 12, the most abundant eluate was detected at the retention time around 1.2 min. The mass spectra at this solution time, shown in Figure S3, demonstrated that they were trimers of propylene imines. The eluate around 4.8 min had the second strongest ion signal (which was also the strongest peak at pH of 8), and other small peaks were also observed around 0.6 min, 2.3 min, and 3.4 min. Their mass spectra were dominated by the protonated ions and ammonium adduct ions of $(C_3H_4O)(C_3H_5N)_3$ and $(C_3H_4O)_2(C_3H_5N)_3$. Based on these observations, we can conclude that oligomers of propylene

imine and acrolein molecules are the major products formed from acrolein with ammonia in the alkaline bulk sample.

At pH=8, though still alkaline, the eluate with the strongest signal intensity was $(C_3H_4O)_2(C_3H_5N)_3$ at retention time around 4.8 min. Trimers of propylene imines were still detected~, but were eluted a little earlier than those in the pH=10 sample.

When the pH decreased to 5-7, the chromatographic traces changed drastically. Pyridinium compound $(C_3H_4O)_2C_6H_8N^+$

became the most abundant eluate, as shown in Figure 6. The $(C_3H_4O)_2C_6H_8N^+$ ion was eluted at 0.8 min in its extracted ion



chromatogram, but the total ions chromatogram showed as a bimodal peak (at 0.7 min and 1.0 min), suggesting that a mixture of other oligomers co-eluted at the same time. The 0.7 min peak had a mixture of $(C_3H_4O)_2C_6H_8N^+$ and trimers of propylene imines, and the 1.0 min peak had two pyridinium compounds $(C_3H_4O)_2C_6H_8N^+$ and $(C_9H_{10}O_2)C_6H_8N^+$ as well as other oligomers. The $(C_9H_{10}O_2)C_6H_8N^+$ here could be the dehydrated form of $(C_3H_4O)_3C_6H_8N^+$. Although polymers

$(C_3H_4O)_m(C_3H_5N)_n$ were still detected, they were not as abundant as in the alkaline solutions, which can be revealed by the decreasing signal intensity at 4.8 min. Hence, neutral or mildly acidic conditions improve the yield of pyridinium compounds in the liquid phase.

For the $(NH_4)_2SO_4$ or $NH_4Cl$ solutions aerosolized into the Tedlar bag the film pH should be 4.6-6 at RH=90-100% according to E-AIM model (http://www.aim.env.uea.ac.uk/aim/aim.php). Therefore, we compared the MS results for the

bag-wall residue sample and for the bulk aqueous reactions at pH=6. Most of the peaks for the bag-wall residue sample were similar to the one in the bulk solution, except for the strongest ion $[(C_3H_4O)_3+NH_4]^+$ at $m/z$ 186.1125. The acrolein trimers were detected in the bag-wall residue but not in the bulk solution because the actual pH in the liquid film in the bag was lower than 6. As we show below, acrolein molecules are prone to oligomerization without reaction with ammonium in the highly acid solutions.

In the solutions with pH=2 and 4, the most abundant eluates were detected at 3.0 min. The corresponding mass spectra were characterized by the strong peak of ammonium adduct ions $[(C_3H_4O)_3+NH_4]^+$ at $m/z$ 186.1125. Other major components were eluted at the retention time around 1.0 min and 4.8 min. However, the formula assignment showed that the main components in these eluates were neither pyridinium salts nor the oligomers of acrolein and propylene imine (Figure S4). In fact, the carbon number for many assigned formulae was not even the multiple of 3, and some ions had very low

DBE values. A possible reason could be that high acidity leads to the acid-catalyzed cleavage reaction of the acrolein and its polymers.

The products observed at various pH values were also examined using SERS and UV/vis adsorption spectroscopy. The Raman spectrum for 3-picoline should be dominated by a band around the 1040 cm$^{-1}$ region (assigned to the $\delta_{ring}$ and $\nu_{ring}$ fundamentals) and a band around the 1630 cm$^{-1}$ region (symmetric ring-stretching), with other characteristic bands with

medium-weak intensity include those around 540 cm$^{-1}$ and 1230 cm$^{-1}$ (Centeno et al., 2012; Guerrini et al., 2009). As shown



in Figure 7, these main characteristic Raman vibrations of 3-picoline have been observed in the Raman spectra at pH=6, but not observed at pH=10. The observation of these frequencies further supports that the assignment of $(C_3H_4O)_2C_6H_8N^+$ at *m/z* 206.1178 to a pyridinium compound. The Raman spectra for the bulk solutions at pH=2 have a larger number of vibrational peaks (e.g., from 800 to 1100 cm$^{-1}$), which indicates a more complicated set of products was formed at high acidity. It is

noticed that the foremost characteristic pyridine vibration around 1230 cm$^{-1}$ was still observed at pH=2, but in the HR-MS analysis, the ions at *m/z* 206.1178 were not present pH=2. This indicates that the products in pH=2 still contain some pyridinium compounds, but $(C_3H_4O)_2C_6H_8N^+$ is no longer a major species.

The UV/vis absorption spectra for the bulk solutions can also reflect the influence of pH upon the liquid phase products, as shown in Figure 8(a). For the solutions at pH=10, the band near 300 nm is consistent with the absorption pattern of

standard acrolein. (see the UV/vis pattern of standard acrolein and standard 3-picoline in Figure S5) The strong absorptivity for the wavelength shorter than 250 nm could result from the π-π* transitions involving conjugated bonds in the oligomers of acrolein and propylene imine. When the pH was adjusted to 6, the UV/vis adsorption pattern was different from that for the alkaline solution. A band around 250-280 nm emerged, which is consistent with the characteristic UV absorption of 3-picoline. Besides, the solution at pH=6 possessed the absorptivity for visible light with wavelength longer than 400 nm. A

weak absorption peak was even found in the band near 430 nm, indicating the formation of more chromophores. However, if the pH value was further decreased to 2, the absorbance at the visible wavelength dropped to almost zero. The possible reason for this drop could be acid-catalyzed removal of the conjugated bonds (which agrees with the UPLC ESI HRMS results). The conversion from acrolein to the chromophores and pyridinium compounds was more distinctly as illustrated by UV/vis spectrometry of the solution at pH=6 with varying reaction times. As shown in Figure 8(b), with the reaction went on,

the band around 250-280 nm corresponding to 3-methyl pyridinium rose to replace the acrolein band around 300 nm as the prominent absorption for wavelength over 250 nm. Meanwhile, the light absorption at the wavelength around 430 nm increased as well, implying the generation of chromophores. The results also suggest that the light-absorbing oligomers and pyridinium compounds could be formed from acrolein in the liquid phase in the time scale of a few hours.

**3.4 Atmospheric implications**





Reactions of carbonyl compounds with ammonia or ammonium to form hemiaminals and imines contribute to the formation of light-absorbing SOA in the atmosphere (De Haan et al., 2011; Hawkins et al., 2018; Lee et al., 2013; Powelson et al., 2014). In this work, we found that acrolein, the smallest α,β-unsaturated aldehyde, has the potential to form light-absorbing heterocyclic NOC, as shown in Figure 9. The conversion from acrolein to pyridinium occurs in a few hours.

Acrolein, though volatile, has a moderate solubility in water (the Henry's constant is ~ 8 mol L$^{-1}$ atm$^{-1}$). Since ammonium is a major component in the atmospheric aerosols, the dissolved acrolein may undergo oligomerization and imine formation to form high molecular weight oligomers of acrolein and propylene imine. If the atmospheric particles are neutral or mildly acidic, 3-picolinium will also be formed in the liquid phase and will continue to react with other carbonyl compounds to generate the higher-MW pyridinium SOA compounds. In other word, the gaseous acrolein could be taken up into the

ammonium-containing aerosols to undergo the reaction to be less volatile chromophores. This reactive uptake of acrolein on aerosols could be a source of particulate light-absorbing NOC.

In addition, acrolein in the gaseous phase can also react with the gaseous ammonia, forming the 3-picoline as a semivolatile organic compound. (The Henry's constant for 3-picoline is ~ 54 mol L$^{-1}$ atm$^{-1}$.) The semi-volatile picoline might partition to the liquid phase of aerosols, and when the liquid phase is neutral or mildly acidic, similar reactions could then

occur to form higher molecular weight pyridinium compounds. These reactions in combination lead to involatile compounds that will come part of SOA. It is possible for the aqueous picolinium cations changed to neutral picolines and then re-volatized to the gas phase once the pH value of the liquid phase increased to be alkaline. However, alkaline condition is less common for aerosols in the atmospheric environment where human lives. Therefore, as a common VOC, acrolein ought to be regarded as a potential SOA-precursor.

Ammonia is one of the most common gaseous compounds in the atmosphere in the populated regions (Chang et al., 2016; Wang et al., 2016; Ye et al., 2011). Ammonium widely exist in the particulate matter in the form of inorganic salts (Hu et al., 2011; Huang et al., 2012; Wang et al., 2016). Acidic aerosol particles have been observed in the ambient conditions (Zhang et al., 2012). The heterocyclic NOC observed in this work might enhance the light absorbing capability (Lee et al., 2013) and the reactive oxygen species generation in cells (Dou et al., 2015). Therefore, secondary reactions of α,β-unsaturated

aldehydes with reduced nitrogen should be taken into account when evaluating climate and health effects of SOA.



## 4. Conclusion

In conclusion, we studied acrolein, as a ubiquitous α,β-unsaturated mono-carbonyl VOC in the atmosphere, for its potential to form SOA in reactions with ammonia or ammonium. Tedlar bag experiments in gas-phase and on the bag walls, as well as bulk aqueous-phase experiments were carried out. We did the MS analysis for the gaseous components and the bag-wall residue samples in the reaction bag and for the bulk solutions. Gaseous 3-picoline was observed in the acrolein-ammonia bag, indicating that acrolein can react with gaseous ammonia to form 3-picoline in the gas phase. Polymers and pyridinium compounds were observed in the liquid film on the wall of acrolein-ammonium bags and in the bulk solutions with ammonium. It demonstrated that, in the ammonium-containing liquid phase, dissolved acrolein can react with ammonium to be 3-picolinium, part of which might re-volatilize back to the gas phase as 3-picoline but mostly undergoes additional reaction to be pyridinium salts in the liquid phase. The reaction rate for the pyridinium formation from acrolein is in the scale of a few hours according to this work. Moreover, the pH value is critical to the liquid products of acrolein and ammonium. The pyridinium compounds are more likely formed when the liquid phase is moderately acidic. Both the polymers of acroleins/propylene imines and the pyridinium compounds can increase the light absorptivity of aerosol particles. Since ammonium widely exist in the atmospheric aerosols, this reactive uptake of acrolein into aerosols should be an important sink for acrolein in the atmosphere. Therefore, our work suggests that acrolein is one of the important building blocks in the formation of reactive nitrogen-containing, light-absorbing, heterocyclic SOA compounds.

## Acknowledgements

This work was supported by the National Natural Science Foundation of China (Nos. 21507010, 91544224, 41775150, 41827804, 91743202). The authors thank Prof. Chris Vanderwal for discussions of the possible chemical mechanisms.



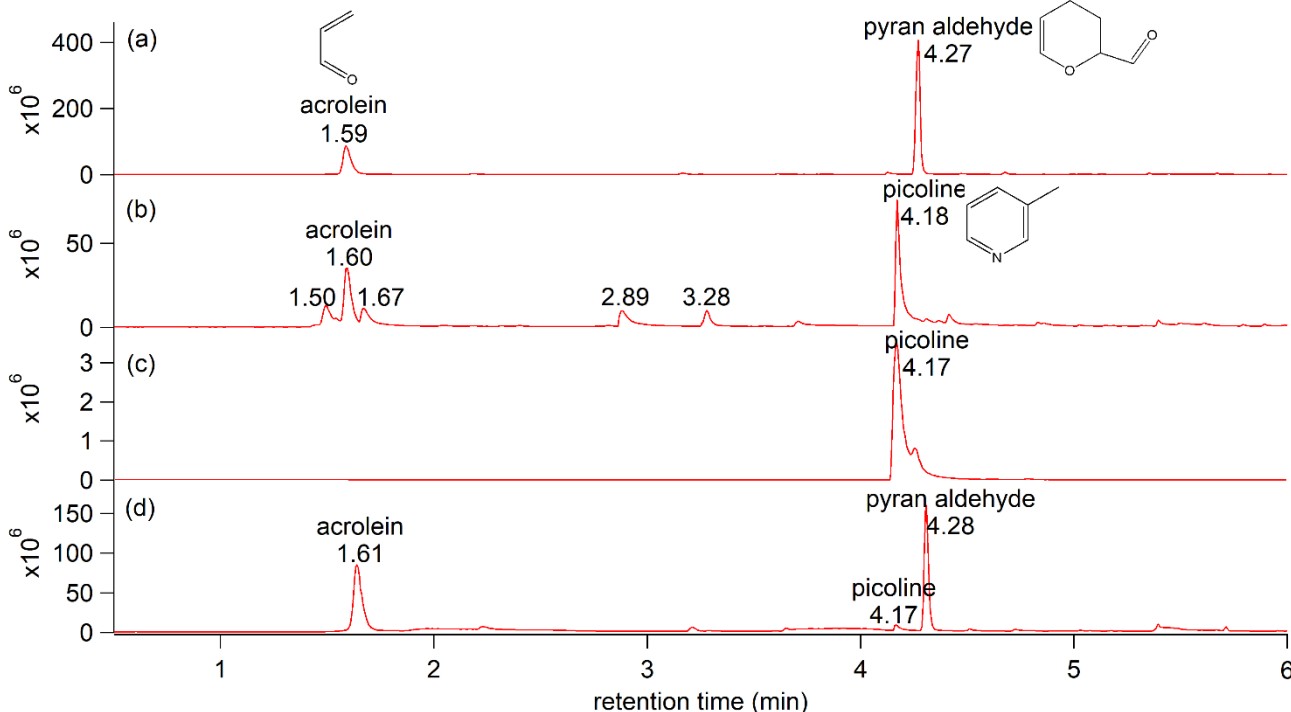

**Figure 1:** **GC traces for: (a) the gaseous components in the control experiment with acrolein only; (b) the gaseous components in the acrolein-ammonia bag; (c) standard 3-picoline; (d) the gaseous components in the acrolein-ammonium bag using $(NH_4)_2SO_4$.**
**The retention times and important compounds identified by NIST library are labelled. The GC-MS results for the**
**acrolein-ammonium experiment using $NH_4Cl$ produced results that were very similar to (d), so they are not included here.**



Reaction 1

Reaction 2

Reaction 3

$C_{12}H_{16}O_2N^+$  206.1176

**Figure 2: Proposed reactions of acrolein with ammonia or ammonium ion that explain the products observed in this work.**



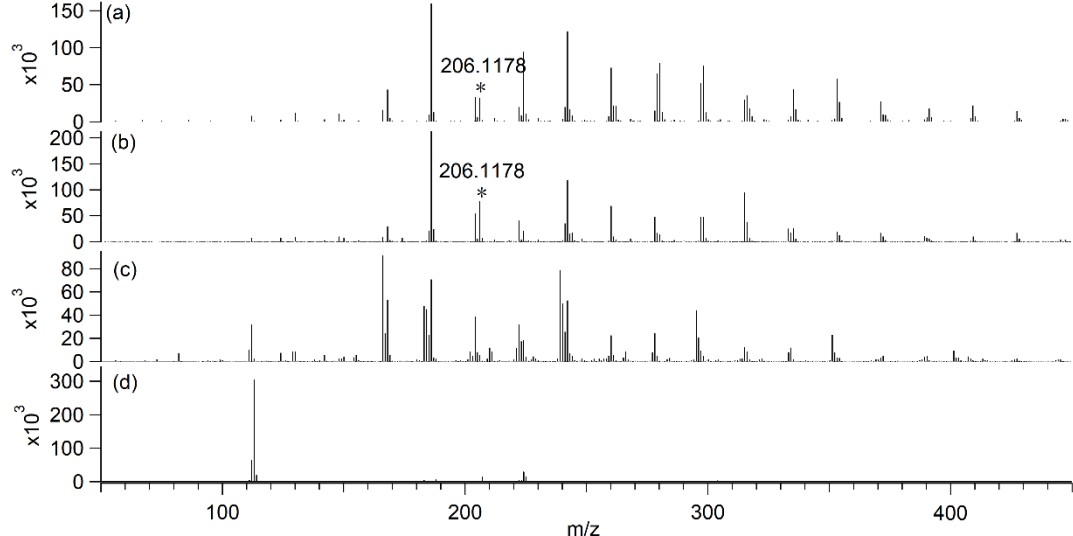

**Figure 3:** The direct-infusion ESI high resolution mass spectra for the bag-wall residue samples in (a) the acrolein-ammonium $NH_4Cl$ bag, (b) the acrolein-ammonium $(NH_4)_2SO_4$ bag and (c) the acrolein-ammonia bag. (d) the mass spectrum for the acrolein standard. The peaks at *m/z* 206.1178 assigned to the pyridinium compounds $(C_3H_4O)_2C_6H_8N^+$ are labelled with asterisks.





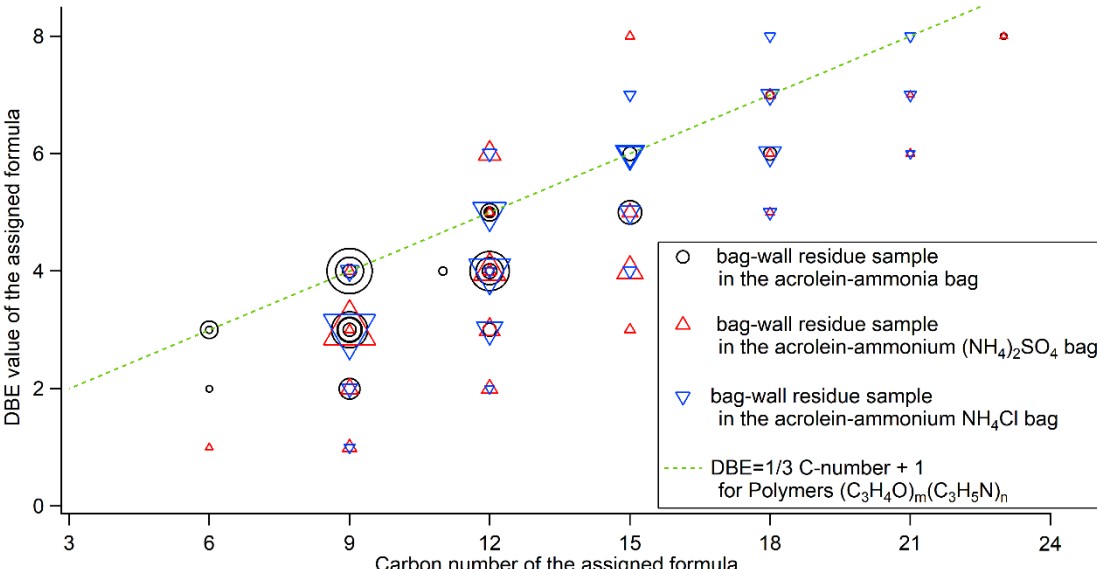

**Figure 4: The DBE values versus the carbon number of the assigned formulae for the top 30 peaks in each mass spectrum in Figure 3. The marker size is proportional to the HRMS peak abundance.**





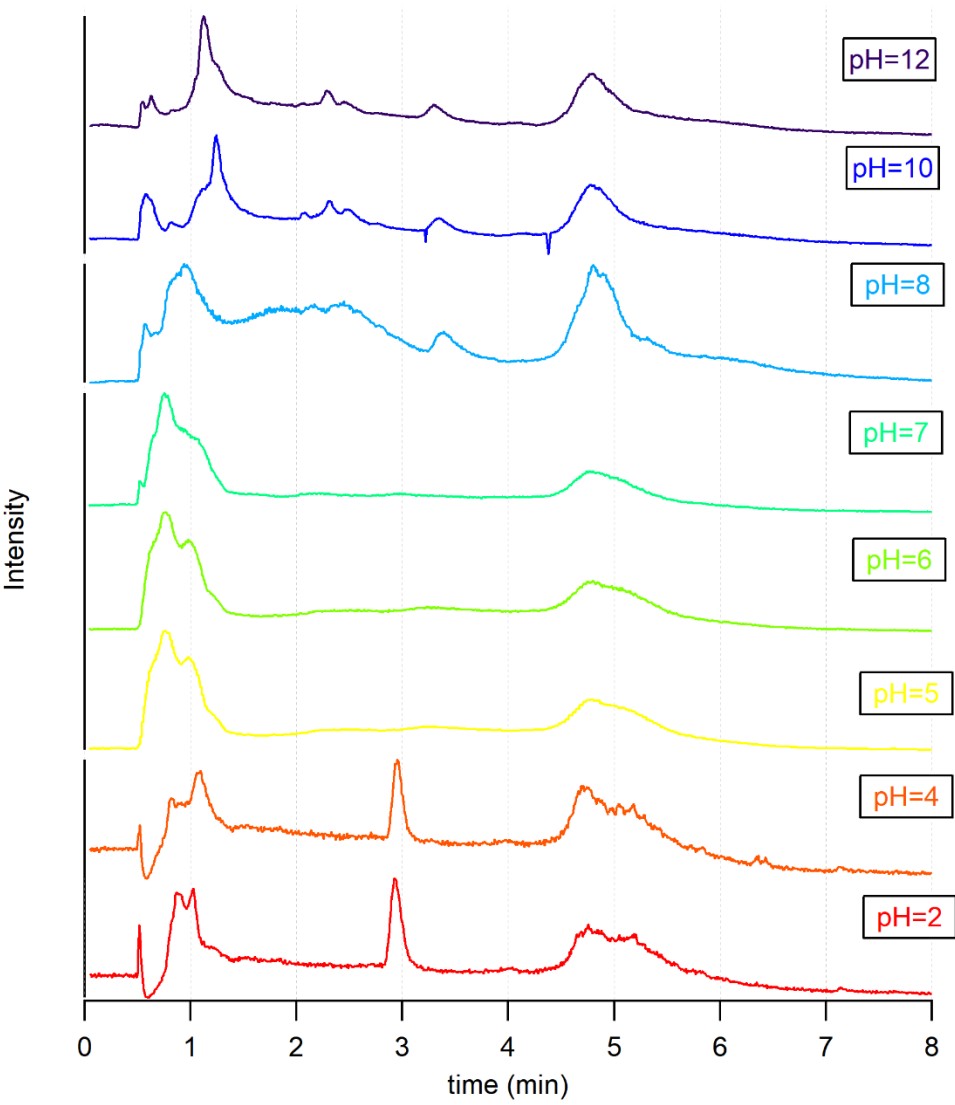

**Figure 5: The total ion chromatographic traces of UPLC for samples in bulk solutions with different pH values.**



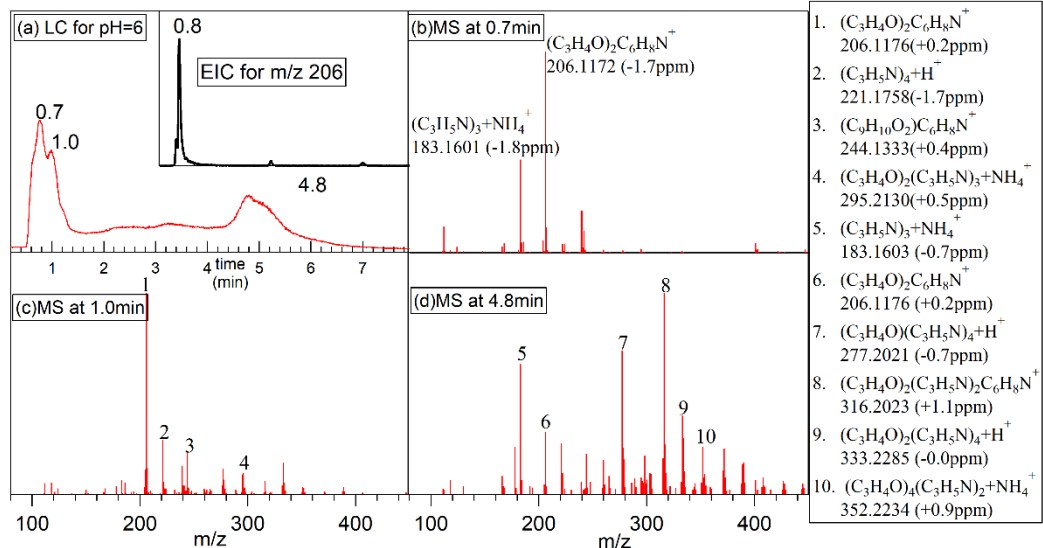

**Figure 6: The UPLC HRMS results for the acrolein reaction with ammonium in the mildly acidic bulk solution. (a) the LC traces for samples in bulk solutions with pH=6. The insert is the extracted ion chromatogram (EIC) for m/z 206. (b-d) the mass spectra for the peaks around 0.7 min, 1.2 min and 4.8 min. The *m/z* values and the corresponding assigned formulae for main ions are marked. The figures in the parentheses are the fractional deviations between the detected and theoretical *m/z* of the assigned ionic formulas.**





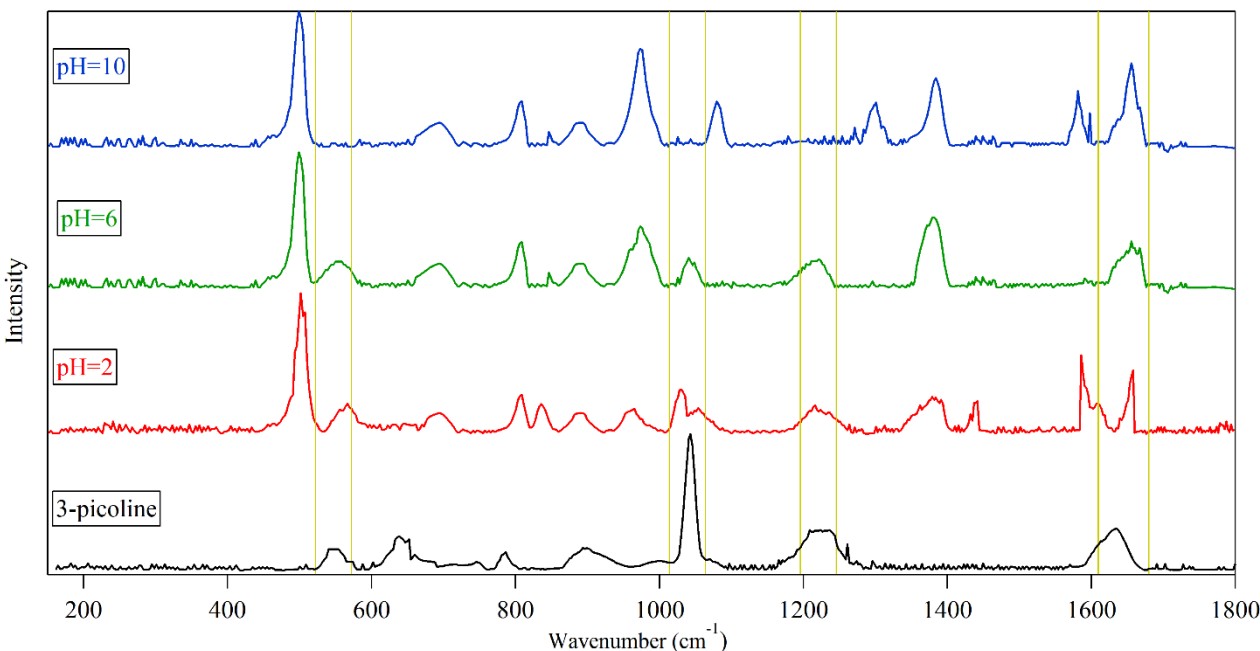

**Figure 7: Surface Enhanced Raman spectra for the bulk solutions with different pH values after a bulk liquid-phase reaction of acrolein with ammonium ion. The bottom panel shows a reference Raman spectrum of 3-picoline.**





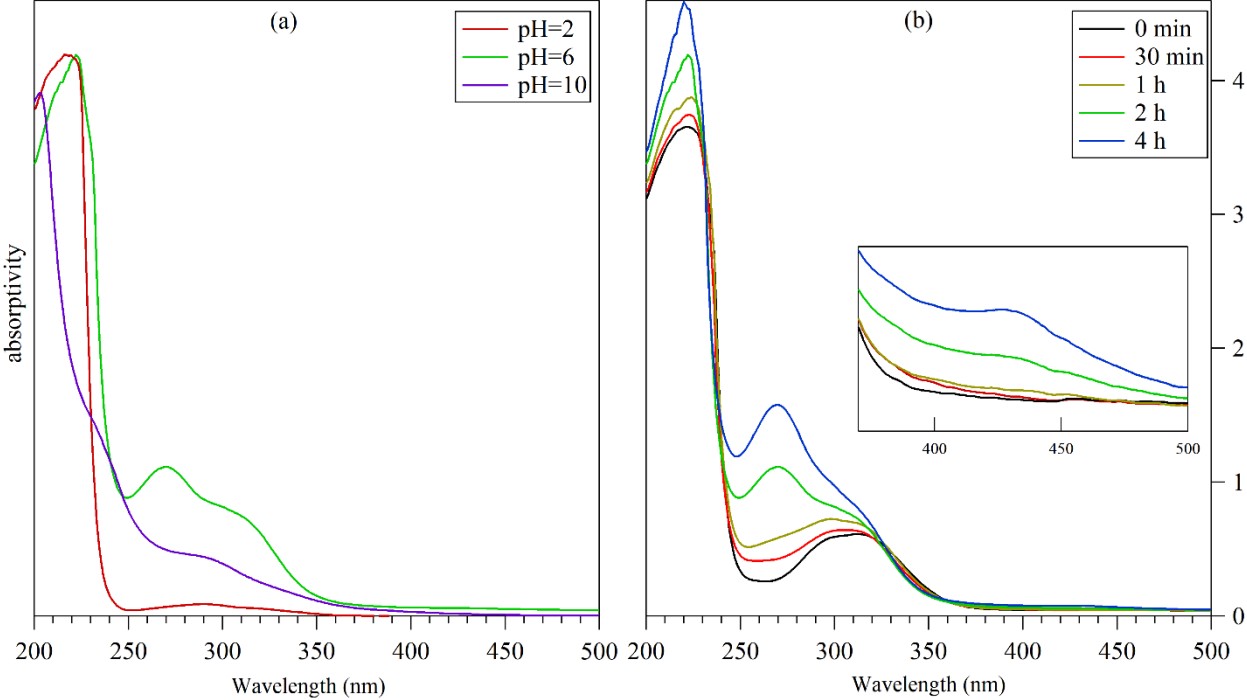

**Figure 8: UV/vis absorption spectra (a) for the bulk solutions with different pH values after a bulk liquid-phase reaction of acrolein with ammonium ion for 2 hours; (b) for the bulk solutions with pH=6 after reaction of acrolein with ammonium ion for different reaction times. The insert in panel (b) is the zoom-in spectra for the wavelength near 430 nm.**



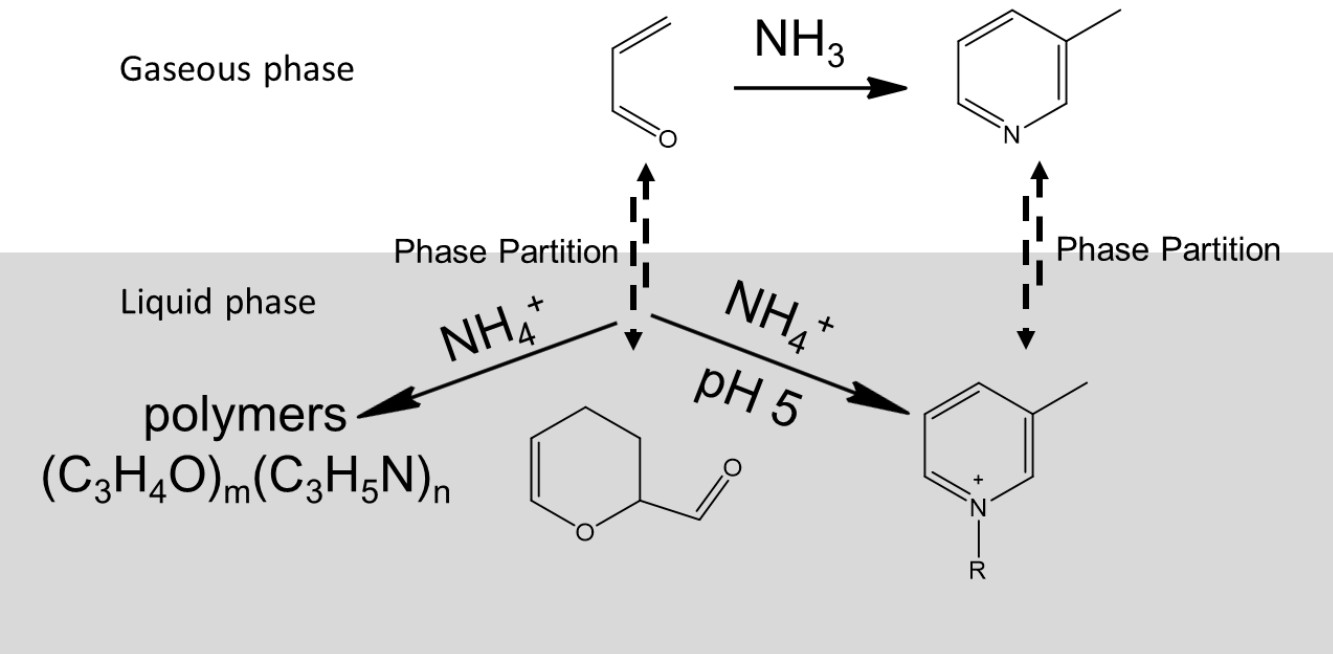

**Figure 9:    Proposed pathways to the formation of SOA compounds by reaction of acrolein with ammonia or ammonium.**



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
