# Peer review of "Nitrogen-containing Secondary Organic Aerosols Formation by Acrolein Reaction with Ammonia/Ammonium"

_Atmospheric Chemistry and Physics, 2018_

## Referee Comment (RC1) · Anonymous Referee #1 · 21 Oct 2018

This manuscript describes laboratory studies of the aqueous aerosol-phase and bulk-phase reactions between acrolein and ammonia / ammonium salts as a function of pH. This work represents a significant advance, as it identifies acrolein for the first time as a precursor for atmospheric brown carbon formation. Furthermore, this work shows that aldehyde + reduced nitrogen reactions appear to be a more general path to brown carbon products than aerosol chemists may currently suppose. This work is publishable after revisions in consideration of the following comments:

One of the conclusions of the paper (repeated in the abstract) is that 3-methylpyrazine may be produced via gas-phase reactions. Both gas-only and aerosol studies were

performed at high humidity in a Tedlar bag, such that condensed water formed on the surface of the bag. As a result, it is challenging to confidently attribute any reaction to gas-phase chemistry alone. In addition, the o-chem literature cited in the paper does not support this, and the authors already provide what seems a more plausible mechanism. It seems that 3-methylpyrazine could be produced in the aqueous phase and then partition to the gas phase if the aqueous phase is alkaline (in non-bulk experiments where this is a possibility).

p. 4 line 4: The authors should also mention here what is known about acrolein photolysis. This might help (or refute) their later argument that reaction with ammonia / ammonium salts is an important sink for acrolein.

p. 5 line 2: How did the RH in the acrolein / ammonia experiment reach 90-100% if dry N2 and no aerosol were added?

Several of the statements made about 3-picoline (for example, p. 8 line 4) would be obvious if the authors would refer to it by its standard name (3-methylpyrazine).

p. 8 last paragraph. The logic of this paragraph is difficult to follow because of the multiple contrasting hypothetical statements (e.g. “should be formed”, “should occur”, “ought to result”). This is one point where pH and protonation effects would seem to explain the partitioning of 3-methylpyrazine.

p. 11: It should be noted that bulk experiments, by preventing opportunities for volatile compounds to evaporate as quickly as they would in aerosol particles, could result in unrealistic over-reactivity of volatile compounds.

p. 12 line 6: The effects described here are similar to how pH influences imidazole formation. Making this connection might assist the reader in developing a general chemical understanding of N-heterocycle formation in clouds.

Bottom of p. 14: It would be helpful to the field if the authors could spell out what additional information, beyond what they have provided, will be necessary in order to

take these reactions into account when evaluating climate or health effects of SOA.

The last two sentences of the conclusion seem premature based on evidence available at this point. In order to demonstrate that reactive uptake into aerosols is an important sink for acrolein, the authors would have to consider all other sinks. As noted earlier, they have not yet considered photolysis, and it is unclear whether they can realistically estimate the uptake rate into aerosol with the information gained in this study in order to compare it with OH oxidation. The last sentence also requires at least semi-quantitative comparison with other sources of N-heterocyclic brown carbon in order to determine if acrolein is “one of the important building blocks.” Thus, these two sentences are not strongly supported by the evidence presented in this work.

---

## Referee Comment (RC2) · Anonymous Referee #2 · 25 Oct 2018

In this work, the authors performed a series of experiments to explore the reaction of acrolein, the smallest $\alpha, \beta$-unsaturated carbonyl, with ammonium and ammonia in both the gas and aqueous phases. Through the use of small chamber and bulk-phase experiments, they showed that acrolein will react with both ammonia in the gas phase and ammonium in the aqueous phase to produce new products, and that the aqueous phase reactions lead to brown carbon formation. This work also identifies several products of these reactions and provides a framework for understanding unsaturated carbonyl reactions in atmospheric water. This work is important for continuing our discussions of the chemistry occurring in these solutions and this manuscript will provide important information to the field after the following comments are addressed.

[Figure]

On page 5, the wording of the statement "The aerosolized $(NH_4)_2SO_4$ or $NH_4Cl$ particles must have quickly deposited to the bag walls forming a coating on the surface" needs to be changed. The statement itself does not seem to fit into this paragraph, but if it stays in the manuscript, the words "must have" needs to be explained or revised.

Due to the fact that the RH in these systems was 90-100%, how can the acrolein/ammonia system be considered to be only a gas phase system? The authors note that a film was formed on the walls, and ammonia and acrolein could easily partition into this film Therefore any compounds observed in this bag could have reacted in the aqueous film and then repartitioned into the gas phase.

In Section 3.1, the authors note that they did not observe pyran aldehyde in the gas phase from the acrolein/ammonia bag. If this dimer was present in the solution before injection, as the authors speculate, then wouldn't it show up in all experiments? Further, if it was formed in the wetted surface of the bag, as also speculated, why doesn't it appear in the presence of ammonium? While the ammonium reaction may be much faster, I would still expect that a small amount of pyran aldehyde might be observed in the gas phase.

Page 8, line 8: Punctuation is needed after the reference at the start of this line.

In the last paragraph of page 8, continuing into page 9, the first sentence is confusing and should be reworded. This paragraph also contains many words such as "should" and "ought" that make it difficult to determine what the authors expected to occur and what actually was observed or deduced.

Page 10, line 22: Do the authors mean "similar to Reaction 2" instead of "similar as Reaction 2?"

Page 11, line 22: There is an odd symbol in the middle of this line.

On page 12, the UPLC results for the acrolein/ammonium bag are compared to the bulk pH=6 solution, but the authors note that the actual pH was somewhere between 4.6

and 6 in the liquid film and that there are acrolein trimers detected in the bag solutions that are not detected in the bulk solutions. They explain that this is because "the actual pH in the liquid film in the bag was lower than 6," but was any comparison done with the bulk solutions with pH=4 or 5? They should work as another point of comparison and give an indication of whether the acidity of the solution was important or if another factor is important here.

Page 14, line 16: "that will come part of SOA" is awkward.

Page 14, line 18: Should "in the atmospheric environment where human lives" be restated as "in the atmospheric environment where humans live" or reworded in some other way?

On page 15, line 12, the authors discuss the pH value being important to the liquid products of this reaction. Are these products really liquid or are the authors referring to products in the aqueous phase?

Throughout the manuscript, there are several grammatical errors and fixing them would improve the readability of the study.
* * *

---

## Referee Comment (RC3) · Anonymous Referee #3 · 20 Nov 2018

The manuscript by Li et al. investigates the formation of organic nitrogen compounds via the reaction of ammonia with acrolein in a series of laboratory experiments. The reaction products are characterized with a suite of analytical instrumentation and the possible contribution of these compounds to light-absorbing organic aerosol (brown carbon; BrC) is investigated through a series of bulk experiments characterized and UV-VIS measurements. The reaction of carbonyl species with ammonia is of interest to the community as it shows that reactions of carbonyls with ammonia may be more general than previously considered. However, I feel that there are several key issues that must be addressed before the manuscript is publishable.

[Figure]

Major comments

1) Gas-phase synthesis of 3-picoline from gaseous acrolein and ammonia seems highly unlikely to me. Although I have not read all of the references provided on pg 7 line 25-pg 8 line 1, those that I am familiar with do not support a gas-phase mechanism. Rather the reactions all require partitioning to some surface. I think it is more likely that this reaction occurs within the aqueous-phase and that 3-picoline then partitions to the gas-phase. On page 8 line 21-22, the authors state that 3-picoline is formed via a gas-phase reaction because no pyridinium compounds were detected in the wall washings. It would be useful to know what the detection limits were for the analytical instruments and what the expected partitioning would be for these compounds. Could it simply be a limit of detection issue?

2) The weak language ("might," "could be," etc.) and the lack of quantitative discussion make the discussion in the Atmospheric Implications section (3.4) unconvincing. The implications need to be considered in a more quantitative manner for the reader to really judge if the process is likely to have an effect. For instance, how does the typical pH of aerosol compare to the results presented in Sect. 3.3 and the implications for BrC? In the current form, this section contains multiple statements that are not supported by the current conclusions. This includes the statement that the conversion occurs in a few hours. While it occurred over the timescale of a few hours in this experiment, if the reaction involves partitioning to the aqueous or condensed phase (which it likely does) the reaction time will depend on environmental factors that have not been thoroughly investigated here. Additionally, the last paragraph of the section is overly simplistic. The last two sentences of the section (reactive oxygen species generation and climate/health effects) should be expanded upon or removed as they are currently not supported by the results.

Other comments

Sect 2.1: Please include the aerosol loading of the experiments.
Sect 2.1: It would be helpful to state the total ammonia (NH3 + NH4) in each experiment.

Page 5 lines 5-6: What exactly is meant by "more than enough water"? How much liquid water is estimated to be in the chamber?

Page 5 lines 7-8: If the aerosol deposited to the wall, this would be observed in the APS data. Was that the case?

Page 5 lines 9-11: "...in order to investigate the gas-phase reaction and liquid-phase reaction respectively." The measurement of a compound in the gas-phase does not necessarily imply that it was formed via a gas-phase mechanism. It could be formed in the condensed-phase and repartition.

Page 7 line 19: Do the authors have a suggestion for why pyran aldehyde was no longer observed?

Page 10 line 22: I would think that the neutral species rather than the cation would be more reactive.

Sect 3.3: This section should include a description of how one expects the chemistry to differ in bulk vs aerosol reactions (i.e., with regards to partitioning of compounds).

Page 11 line 6: I believe the Jang et al. (2003) reference looked at acid-catalyzed reactions of carbonyls. I don't think that reactions of the carbonyls with ammonia were considered.

Page 12 lines 4-6: I don't necessarily see from the figure the decreasing signal intensity at 4.8 min. It appears that it decreases relative to the other peaks, but without a scale I don't know how it relates in terms of absolute signal.

Page 12 line 10: Why not compare to pH 5?

Figure 5: It would be helpful to label the peaks (e.g., label the peak at 1.2 min as propylene imines).

Page 14 line 3-4: "...potential to form light-absorbing..." the pH of aerosols and droplets needs to be discussed before this conclusion can be made.

Page 14 line 12: Please see major comment 1.

Page 15 line 16-17: This sentence is currently unsupported by the atmospheric implications section. It should be removed or the atmospheric implications section should be substantially expanded and made more quantitative.

Figure 9: I don't find this figure particularly helpful. The reaction pathways have already been discussed.

General: There are numerous grammatical errors, many dealing with use of definite articles and subject-verb agreement. Correction of these errors would improve readability of the paper.

General: The resolution of the figures should be improved.

---

## Author Comment (AC1) · 27 Dec 2018

**Dear Editors and Reviewers:**

We thank the editors and the reviewers for your thoughtful and valuable comments. We have numbered the comments and reproduced them below in black. Our point-to-point responses were also written below, but in blue. Sentences cited from the revised manuscript were ***in black and in bold, italic type***. Changes made to the manuscript are marked in red in the submitted revision file.

Referee 1

This manuscript describes laboratory studies of the aqueous aerosol-phase and bulkphase reactions between acrolein and ammonia / ammonium salts as a function of pH. This work represents a significant advance, as it identifies acrolein for the first time as a precursor for atmospheric brown carbon formation. Furthermore, this work shows that aldehyde + reduced nitrogen reactions appear to be a more general path to brown carbon products than aerosol chemists may currently suppose. This work is publishable after revisions in consideration of the following comments:

(1) One of the conclusions of the paper (repeated in the abstract) is that 3-methylpyrazine may be produced via gas-phase reactions. Both gas-only and aerosol studies were performed at high humidity in a Tedlar bag, such that condensed water formed on the surface of the bag. As a result, it is challenging to confidently attribute any reaction to gas-phase chemistry alone. In addition, the o-chem literature cited in the paper does not support this, and the authors already provide what seems a more plausible mechanism. It seems that 3-methylpyrazine could be produced in the aqueous phase and then partition to the gas phase if the aqueous phase is alkaline (in non-bulk experiments where this is a possibility).

The question whether it is a direct gas-phase reaction or a liquid-phase reaction followed by evaporation is hard to answer, and we spent quite a bit of time thinking about this issue.

In the original manuscript, the last paragraph in Page 8 aimed to explain our initial thoughts. In that paragraph, we mentioned that the observed 3-picoline (we used its standard name 3-methylpyridine in the revised version) in ammonium system was consistent with liquid phase reactions followed by the liquid-to-gas partitioning, because pyridinium products were observed in the film on the bag wall. On the other hand, for the 3-picoline in the ammonia system, a direct gas-phase reaction was suggested as the main pathway because of the lack of detection of pyridinium compounds either in the liquid film of bag wall or in the bulk experiments for the pH=10 solution.

During this revision, we re-assessed our initial conclusions based on the comments by you and other referees. We have now realized that no detection of pyridinium compounds in the liquid film for the ammonia system can result from the high-pH effect. The pH value for the liquid film for the ammonia system was probably higher than 10. After the pyridine ring products form in the liquid phase, due to the aqueous equilibrium of $C_6H_7N$ and $C_6H_8N^+$ and the pKa of ~5.68, they tend to be neutral molecules of $C_6H_7N$ under the alkaline condition. The $C_6H_7N$ are semi-volatile (Henry's constant of ~54 mol $L^{-1}$ $atm^{-1}$), so they will further evaporate into the gas phase. This would explain why large amounts of gaseous 3-methylpyridine was observed by GC-MS while no pyridinium compounds were detected in aqueous samples for the ammonia system. In addition, the bulk aqueous solution experiments demonstrated that the acrolein in the higher alkaline solution was more likely to generate the oligomers rather than the heterocyclic compounds. We should note that while the liquid-to-gas partitioning of liquid-phase products is the most likely sources of the observed gaseous 3-methylpyridine, the data are not sufficient to completely rule

out the possibility of a direct gas-phase reaction.

Therefore, in the revised manuscript, Reaction 2 was mentioned as a liquid-phase reaction, and we emphasized the liquid-to-gas partitioning of the liquid-phase products as the source of the gaseous 3-methylpyridine observed in both ammonium system and ammonia system. (But direct Reaction 2 in the gas phase was not absolutely ruled out.) We have made a revision for the last paragraph of Page 8 and for the first paragraph of Section 3.3. The original sentences or phrases about direct gas-phase reaction to form 3-picoline have been deleted or re-written as well.

Revised Paragraph from Line 22, Page 8 to Line 4, Page 9: *The design of the experiments makes it challenging to determine whether the proposed reaction is occurring in the gas phase or in the liquid phase. The gaseous 3-methylpyridine was detected in both the acrolein-ammonium bag and the acrolein-ammonium bag. In both types of experiments, gaseous ammonium was present in the gas phase and ammonium ions were present in the liquid film on the bag walls. An important clue is that, in addition to the observed gaseous 3-methylpyridine, the pyridinium products were observed in the liquid film on the wall of the acrolein-ammonium bag (see section 3.2). Hence, we propose that Reaction 2 occurred in the liquid phase in our experiments (although we cannot rule out a gas-phase reaction.) The gaseous 3-methylpyridine result from liquid-to-gas partitioning equilibrium promoted by the limited solubility of 3-methylpyridine (its Henry's constant is ~ 54 mol $L^{-1}$ $atm^{-1}$).*

Revised Lines 7-14, Page 11: *The difference in the aqueous film pH values is the possible cause. The formation of imidazole chromophores and other light-absorbing NOC in reactions involving ammonia has been demonstrated to be pH-dependent by several reports (Kampf et al., 2012; Phillips et al., 2017; Yu et al., 2011). In the Tedlar-bag experiments, the liquid film on the acrolein-ammonia bag-wall was alkaline. As the pKa of 3-methylpyridinium is 5.68 (at 20 °C), under the alkaline condition, the products of Reaction 2 in the liquid film predominantly exist as the neutral molecules of 3-methylpyridine rather than 3-methylpyridinium cations, and more easily volatilize to the gas phase. This explains the observation of large levels of 3-methylpyridine by GC-MS but no 3-methylpyridinium cations detectable by ESI HRMS for the acrolein-ammonia bag.*

Revised title of section 3.1: *Analysis of gaseous components in the acrolein-ammonia and acrolein-ammonium bags*

(2) p. 4 line 4: The authors should also mention here what is known about acrolein photolysis. This might help (or refute) their later argument that reaction with ammonia /ammonium salts is an important sink for acrolein.

In the revised introduction, we have cited Magneron et al. (2002) work on the photolysis and radical reaction of acrolein. Typical atmospheric lifetime of acrolein with respect to reaction with OH is a few hours, and with respect to photolysis is several days. In our work, the reaction experiments were carried out ~2 hours, which is comparable to the acrolein lifetime with respect to its reaction with OH. Hence the several-hour reaction of acrolein with ammonia can reasonably compete with reaction with OH, at least under the concentration conditions used in our experiments. In addition, we have compared the Tedlar bag experiments using dark condition and under ambient sunlight condition (this part of work is not mentioned in the manuscript), the MS results were very similar, furtherly indicating the photolysis of acrolein is not fast. The resulting revisions were:

Revised Lines 1-2, Page 4: *The atmospheric lifetime of acrolein is >6 days towards photolysis, and is few hours towards OH radical reaction (Magneron et al., 2002).*

Revised Lines 19-23, Page 14: *The conversion from acrolein to pyridinium compounds occurs on a time scale of a few hours. This conversion time is comparable to that of the reaction of acrolein with OH, and much shorter than that of the acrolein photolysis (Magneron et al., 2002). Therefore, aqueous removal of acrolein could compete with its gas phase oxidation. This assertion will need to be verified in future studies under the conditions of the more atmospherically relevant acrolein and ammonia concentrations.*

(3) p. 5 line 2: How did the RH in the acrolein / ammonia experiment reach 90-100% if dry N2 and no aerosol were added?

The following sentence was shown on Lines 19-21, Page 4 in the revised manuscript, "*By evaporating 2 mL of the acrolein solution and 400 μL of the ammonium hydroxide solution in the water bath at 40°C, nearly 5 mmol total ammonia and 0.3 mmol acrolein were introduced into the acrolein-ammonia bag*". Though no ammonium aerosols were added into the bag system, about 2.4 mL of water was also added into the ammonia-system bag. Hence the RH for the ammonia bag reached as high as that for the ammonium bags. We have added a phrase in the revised Lines 9-11, Page 5: "*The RH in the acrolein-ammonia experiments was also high, around 90%, because ~2 g $H_2O$ vapor was also added to the bag when evaporating 2 mL of the acrolein solution and 400 μL of the ammonium hydroxide solution.*"

(4) Several of the statements made about 3-picoline (for example, p. 8 line 4) would be obvious if the authors would refer to it by its standard name (3-methylpyrazine).

We changed the "picoline" into its standard name "methylpyridine" in the revised manuscript.

(5) p. 8 last paragraph. The logic of this paragraph is difficult to follow because of the multiple contrasting hypothetical statements (e.g. "should be formed", "should occur", "ought to result"). This is one point where pH and protonation effects would seem to explain the partitioning of 3-methylpyrazine.

As answered to your first comment, we have rewritten this paragraph and increased the level of certainty in our discussion. In the revised manuscript, liquid-to-gas partitioning, rather than the direct gas-phase reaction, was proposed as the source of the GC-MS observed gaseous 3-picoline. pH and protonation effects explain the liquid-to-gas partitioning of picoline for the ammonia system. We further discuss the pH effects it in the beginning of Section 3.3 where pH is discussed.

Revised Last Paragraph of Page 8: *The design of the experiments makes it challenging to determine whether the proposed reaction is occurring in the gas phase or in the liquid phase. The gaseous 3-methylpyridine was detected in both the acrolein-ammonium bag and the acrolein-ammonium bag. In both types of experiments, gaseous ammonium was present in the gas phase and ammonium ions were present in the liquid film on the bag walls. An important clue is that, in addition to the observed gaseous 3-methylpyridine, the pyridinium products were observed in the liquid film on the wall of the acrolein-ammonium bag (see section 3.2). Hence, we propose that Reaction 2 occurred in the liquid phase in our experiments (although we cannot rule out a gas-phase reaction.) The gaseous 3-methylpyridine result from liquid-to-gas partitioning equilibrium promoted by the limited solubility of 3-methylpyridine (its Henry's*

*constant is ~ 54 mol L$^{-1}$ atm$^{-1}$).*

Revised Lines 7-14, Page 11: ***The difference in the aqueous film pH values is the possible cause. The formation of imidazole chromophores and other light-absorbing NOC in reactions involving ammonia has been demonstrated to be pH-dependent by several reports (Kampf et al., 2012; Phillips et al., 2017; Yu et al., 2011). In the Tedlar-bag experiments, the liquid film on the acrolein-ammonia bag-wall was alkaline. As the pKa of 3-methylpyridinium is 5.68 (at 20 °C), under the alkaline condition, the products of Reaction 2 in the liquid film predominantly exist as the neutral molecules of 3-methylpyridine rather than 3-methylpyridinium cations, and more easily volatilize to the gas phase. This explains the observation of large levels of 3-methylpyridine by GC-MS but no 3-methylpyridinium cations detectable by ESI HRMS for the acrolein-ammonia bag.***

(6) p. 11: It should be noted that bulk experiments, by preventing opportunities for volatile compounds to evaporate as quickly as they would in aerosol particles, could result in unrealistic over-reactivity of volatile compounds.

Thanks for this comments. We have mentioned this in the revised manuscript, and it also helped explain some difference between the bag-wall sample and bulk sample. We avoided calling it "over-reaction" but instead referred to it as a "more complete reaction".

Revised Lines 15-20, Page 12: ***This difference could be due to the more complete reaction in the bulk experiments. Since the bulk solution was sealed in a bottle with a relatively small head-space to undergo the aqueous-phase reactions, the volatile or semi-volatile products or intermediate products could not evaporate into the gas phase as sufficiently as they could in the Tedlar bag experiments. This enhanced the contribution of volatile and semi-volatile compounds to the reaction, resulting in the observed difference between the bag-wall residue sample and the bulk sample.***

(7) p. 12 line 6: The effects described here are similar to how pH influences imidazole formation. Making this connection might assist the reader in developing a general chemical understanding of N-heterocycle formation in clouds.

In the revised Section 3.3, we have mentioned the known pH effects on the heterocyclic NOC formation. And just like the light-absorbing imidazole formation is decreased by decreasing pH value, the similar reason explains the reduced formation of pyridinium compounds when the bulk solution is acidic.

Revised Lines 7-9, Page 11: ***The difference in the aqueous film pH values is the possible cause. The formation of imidazole chromophores and other light-absorbing NOC in reactions involving ammonia has been demonstrated to be pH-dependent by several reports (Kampf et al., 2012; Phillips et al., 2017; Yu et al., 2011).***

Revised Sentences from Line 25, Page 12 to Line 3, Page 13: ***In other word, the reaction of acrolein with ammonia/ammonium was inhibited in the highly acidic solutions. A similar observation was made for other heterocyclic NOC produced by carbonyl-to-imine conversion (Kampf et al., 2012). The formation of imine from the carbonyl requires the free electron pair on ammonia, which is reduced in abundance under in an acidic environment due to the pH dependent equilibrium of ammonium ions and ammonia.***

(8) Bottom of p. 14: It would be helpful to the field if the authors could spell out what additional information, beyond what they have provided, will be necessary in order to take these reactions into account when evaluating climate or health effects of SOA.

We have rewritten the Section 3.4, and deleted some statements not supported by the evidence presented in this work. In addition, we mentioned our mechanism will need to be verified in future studies under the conditions of the more atmospherically relevant acrolein and ammonia concentrations

(9) The last two sentences of the conclusion seem premature based on evidence available at this point. In order to demonstrate that reactive uptake into aerosols is an important sink for acrolein, the authors would have to consider all other sinks. As noted earlier, they have not yet considered photolysis, and it is unclear whether they can realistically estimate the uptake rate into aerosol with the information gained in this study in order to compare it with OH oxidation. The last sentence also requires at least semi-quantitative comparison with other sources of N-heterocyclic brown carbon in order to determine if acrolein is "one of the important building blocks." Thus, these two sentences are not strongly supported by the evidence presented in this work.

We agree that the original last two sentences of the conclusion were not fully supported by the presented evidence. We have revised the last sentences of the conclusion, and deleted the phrase "the important sink" and "the important building blocks".

Revised Lines 22-24, Page 15: ***Since ammonium widely exists in the atmospheric aerosols, our work suggests that the reactive uptake of acrolein into aerosols is a potential atmospheric loss process for acrolein, which leads to the formation of nitrogen-containing, light-absorbing, heterocyclic SOA compounds.***

We attempted to provide a more quantitative description of the reaction rate by analyzing bulk solution experiments at different reaction times. However, it was difficult to quantitatively measure the concentration of acrolein in the solution, because the ESI MS only observes its dimer's protonated ions. As shown in the left figure below, the dimer's signal did not decrease obviously during the reaction. We also tried to analyze the rate of formation of pyridinium products, such as $(C_3H_4O)_2C_6H_8N^+$. As shown in the right figure below, the signal increased on a time scale of hours. However, the lack of the standard for $(C_3H_4O)_2C_6H_8N^+$ made the quantitative analysis infeasible. Therefore, we elected to leave the discussion qualitative at this point.

[Figure]

time variation of the peak area
for m/z 113 (acrolein dimer)

[Figure]

time variation of the peak area
for m/z 206 (product of $(C_3H_4O)_2C_6H_8N^+$)

Referee 2

In this work, the authors performed a series of experiments to explore the reaction of acrolein, the smallest α,β-unsaturated carbonyl, with ammonium and ammonia in both the gas and aqueous phases. Through the use of small chamber and bulk-phase experiments, they showed that acrolein will react with both ammonia in the gas phase and ammonium in the aqueous phase to produce new products, and that the aqueous phase reactions lead to brown carbon formation. This work also identifies several products of these reactions and provides a framework for understanding unsaturated carbonyl reactions in atmospheric water. This work is important for continuing our discussions of the chemistry occurring in these solutions and this manuscript will provide important information to the field after the following comments are addressed.

(1) On page 5, the wording of the statement "The aerosolized (NH$_4$)$_2$SO$_4$ or NH$_4$Cl particles must have quickly deposited to the bag walls forming a coating on the surface" needs to be changed. The statement itself does not seem to fit into this paragraph, but if it stays in the manuscript, the words "must have" needs to be explained or revised.

We have revised this sentence in the revised manuscript.

Revised Line 6, Page 5: ***The aerosolized (NH$_4$)$_2$SO$_4$ or NH$_4$Cl particles can be expected to deposit on the bag walls, at least partially, after some time.***

(2) Due to the fact that the RH in these systems was 90-100%, how can the acrolein/ammonia system be considered to be only a gas phase system? The authors note that a film was formed on the walls, and ammonia and acrolein could easily partition into this film Therefore any compounds observed in this bag could have reacted in the aqueous film and then repartitioned into the gas phase.

Thank you for this useful comment. We agree that the acrolein-ammonia system cannot be simply considered as a gas-phase system. We have changed the title of section 3.1 to be "***Analysis of gaseous components in the acrolein-ammonia and acrolein-ammonium bags***". In the revised manuscript, Reaction 2 was mentioned as a liquid-phase reaction, and we emphasized the liquid-to-gas partitioning of the liquid-phase products as the source of the gaseous 3-methylpyridine observed in both ammonium system and ammonia system. (But direct Reaction 2 in the gas phase was not absolutely ruled out.) We have made a revision for the last paragraph of Page 8 and for the first paragraph of Section 3.3. The original sentences or phrases about direct gas-phase reaction to form 3-picoline have been deleted or re-written as well. The more details were described in our response to comment (1) of Referee 1.

(3) In Section 3.1, the authors note that they did not observe pyran aldehyde in the gas phase from the acrolein/ammonia bag. If this dimer was present in the solution before injection, as the authors speculate, then wouldn't it show up in all experiments? Further, if it was formed in the wetted surface of the bag, as also speculated, why doesn't it appear in the presence of ammonium? While the ammonium reaction may be much faster, I would still expect that a small amount of pyran aldehyde might be observed in the gas phase.

We have carefully checked the GC-MS results for the acrolein-ammonia bag (Figure 1b). There is a very small GC peak (overwhelmed by the neighbor 3-picoline peak, unable to be

auto-recognized by the GC-MS analyzing system) corresponding to the dimer. So, as you suggested, there was still a very small amount of pyran aldehyde in the gas phase in the ammonium system. We made it clearer in the revised Line 19, Page 7: ***Pyran aldehyde was still observed, but as an inconspicuous peak, as shown in Figure 1(b).***

(4) Page 8, line 8: Punctuation is needed after the reference at the start of this line.
Corrected in the revised Line 9, Page 8.

(5) In the last paragraph of page 8, continuing into page 9, the first sentence is confusing and should be reworded. This paragraph also contains many words such as "should" and "ought" that make it difficult to determine what the authors expected to occur and what actually was observed or deduced.
We have rewritten this paragraph and increased the level of certainty in our discussion. Please refer to our response to comment (5) of Referee 1.

(6) Page 10, line 22: Do the authors mean "similar to Reaction 2" instead of "similar as Reaction 2?"
In the original manuscript, Reaction 2 was considered as the direct gas-phase reaction to form gaseous 3-methylpyridine, so we used "similar to Reaction 2" when we mentioned the liquid-phase reaction. In the revised manuscript, since the gaseous 3-methylpyridine was attributed to the phase partitioning of the liquid-phase product, we rewritten this sentence without the word "similar".
Revised Lines 18-21, Page 10: ***We propose the following pathway to $(C_3H_4O)_2C_6H_8N^+$: Reaction 2 in Figure 2 leads to the formation of 3-methylpyridine, and then the lone pair on the pyridine nitrogen can attack the electrophilic site in the carbonyl of the acrolein dimer, to change the carbonyl to hemiaminal and form pyridinium compounds $(C_3H_4O)_2C_6H_8N^+$ (Reaction 3 in Figure 2).***

(7) Page 11, line 22: There is an odd symbol in the middle of this line.
Thank you. We have deleted this typo in the revised version.

(8) On page 12, the UPLC results for the acrolein/ammonium bag are compared to the bulk pH=6 solution, but the authors note that the actual pH was somewhere between 4.6 and 6 in the liquid film and that there are acrolein trimers detected in the bag solutions that are not detected in the bulk solutions. They explain that this is because "the actual pH in the liquid film in the bag was lower than 6," but was any comparison done with the bulk solutions with pH=4 or 5? They should work as another point of comparison and give an indication of whether the acidity of the solution was important or if another factor is important here.
We agree with this comment. We cannot attribute the difference between bag-wall sample and bulk solution simply to the pH value. In the revised manuscript, we pointed out that the possible more complete reaction of some volatile products in the bulk, which could not evaporate into the gas phase as sufficiently as they could in the bag experiments.
Revised Lines 15-20, Page 12: ***This difference could be due to the more complete reaction in the***

*bulk experiments. Since the bulk solution was sealed in a bottle with a relatively small head-space to undergo the aqueous-phase reactions, the volatile or semi-volatile products or intermediate products could not evaporate into the gas phase as sufficiently as they could in the Tedlar bag experiments. This enhanced the contribution of volatile and semi-volatile compounds to the reaction, resulting in the observed difference between the bag-wall residue sample and the bulk sample.*

(9) Page 14, line 16: "that will come part of SOA" is awkward.
Since we have rewritten the Section 3.4, this awkward phrase has been deleted.

(10) Page 14, line 18: Should "in the atmospheric environment where human lives" be restated as "in the atmospheric environment where humans live" or reworded in some other way?
We removed this segment.
Revised Line 4, Page 15: *alkaline conditions are much less common*

(11) On page 15, line 12, the authors discuss the pH value being important to the liquid products of this reaction. Are these products really liquid or are the authors referring to products in the aqueous phase?
Thank you for noticing this. The products are not always liquid. Like 3-methylpyridine, which is the important intermediate products in the aqueous phase, it is semi-volatile so it can go to the gas phase.
In the revised manuscript, we used the more precise words. The original sentence "the pH value is critical to the liquid products of acrolein and ammonium" has been revised to "*Moreover, the pH value is critical to the reaction of acrolein and ammonium in the aqueous phase*." in the revised Lines 19-20, Page 15. In addition, in the abstract and conclusion sections, we have used "aqueous phase" to replace the original "liquid phase". We did not replace them all through the paper, since in our bulk experiments, the liquid phase is the aqueous phase, and there is no ambiguity.

(12) Throughout the manuscript, there are several grammatical errors and fixing them would improve the readability of the study.
We have corrected as many grammatical errors and typos as we could. We hope the rest will be caught at the copy-editing stage.

Referee 3

The manuscript by Li et al. investigates the formation of organic nitrogen compounds via the reaction of ammonia with acrolein in a series of laboratory experiments. The reaction products are characterized with a suite of analytical instrumentation and the possible contribution of these compounds to light-absorbing organic aerosol (brown carbon; BrC) is investigated through a series of bulk experiments characterized and UV-VIS measurements. The reaction of carbonyl species with ammonia is of interest to the community as it shows that reactions of carbonyls with ammonia may be more general than previously considered. However, I feel that there are several key issues that must be addressed before the manuscript is publishable.

Major comments

(1) Gas-phase synthesis of 3-picoline from gaseous acrolein and ammonia seems highly unlikely to me. Although I have not read all of the references provided on pg 7 line 25-pg 8 line 1, those that I am familiar with do not support a gas-phase mechanism. Rather the reactions all require partitioning to some surface. I think it is more likely that this reaction occurs within the aqueous-phase and that 3-picoline then partitions to the gas-phase. On page 8 line 21-22, the authors state that 3-picoline is formed via a gas-phase reaction because no pyridinium compounds were detected in the wall washings. It would be useful to know what the detection limits were for the analytical instruments and what the expected partitioning would be for these compounds. Could it simply be a limit of detection issue?

Our initial thoughts for the 3-picoline formation in the original manuscript are that a direct gas-phase reaction was suggested as the main pathway because of the lack of detection of pyridinium compounds either in the liquid film of acrolein-ammonia bag wall or in the bulk experiments for the pH=10 solution.

During this revision, we re-assessed our initial conclusions and have now realized that no detection of pyridinium compounds in the liquid film for the ammonia system can result from the high-pH effect. (The detection limits you mentioned in this comment could be also one of the reason, but we don't think it the main cause.)

Therefore, in the revised manuscript, Reaction 2 was mentioned as a liquid-phase reaction, and we emphasized the liquid-to-gas partitioning of the liquid-phase products as the source of the gaseous 3-methylpyridine observed in both ammonium system and ammonia system. But direct Reaction 2 in the gas phase was not absolutely ruled out. We have made a revision for the last paragraph of Page 8 and for the first paragraph of Section 3.3. The original sentences or phrases about direct gas-phase reaction to form 3-picoline have been deleted or re-written as well.

Since similar concerns were also raised by the other two reviewers, we have given a detailed explanation about this concern in our response to the comment (1) of Referee 1. Please see that response for more details.

(2) The weak language ("might," "could be," etc.) and the lack of quantitative discussion make the discussion in the Atmospheric Implications section (3.4) unconvincing. The implications need to be considered in a more quantitative manner for the reader to really judge if the process is likely to have an effect. For instance, how does the typical pH of aerosol compare to the results presented in Sect. 3.3 and the implications for BrC? In the current form, this section contains multiple statements that are not supported by the current conclusions. This includes the statement that the

conversion occurs in a few hours. While it occurred over the timescale of a few hours in this experiment, if the reaction involves partitioning to the aqueous or condensed phase (which it likely does) the reaction time will depend on environmental factors that have not been thoroughly investigated here. Additionally, the last paragraph of the section is overly simplistic. The last two sentences of the section (reactive oxygen species generation and climate/health effects) should be expanded upon or removed as they are currently not supported by the results.

We have rewritten the section 3.4. We avoided the ambiguous statements as much as possible, and deleted some statements not supported by the evidence presented in this work (such as the ROS and climate related contents). In the revised section 3.4, we focused on the acrolein-to-pyridinium pathway as a potential source of light-absorbing NOC in SOA. Also, we mentioned our mechanism will need to be verified in future studies under the conditions of the more atmospherically relevant acrolein and ammonia concentrations

Our experiments show that acrolein-to-pyridinium pathway is likely to occur under the moderately acidic conditions, which is relevant for aerosol particles and fog droplets. For example, it has potential importance of this NOC formation pathway because aerosols in winter haze of northern China cities have been reported as moderately acidic.

The reaction occurs on a time scale of a few hours according to our experiments. Though we do not have detailed kinetics data, we can roughly compare the reaction time with the atmospheric lifetime of acrolein with respect to photolysis and OH radical reaction. The comparison suggests that the 2-hour reaction (under our experimental conditions) makes it possible for the acrolein in the atmosphere to react with ammonia before acrolein is consumed by photolysis or OH reactions. In addition, we have tried to do a more quantitative analysis for the reaction rate, as described in our response to comment (9) of referee 1. However, the lack of direct measurements of acrolein concentrations in the liquid phase makes more quantitative description of the system difficult.

Other comments

(3) Sect 2.1: Please include the aerosol loading of the experiments.

After we atomized the ammoniums into the bag, some of the the aerosols would be suspended in the bag and some would be deposited on the bag-wall. In a hypothetical situation with no wall deposition, the ammonium aerosol loading would be about 2000 mg/m$^3$. Since it is a very high concentration, most of the particles should quickly coagulate and deposit on the wall, so the calculated loading is not meaningful. Therefore, we did not calculate the mass concentration of aerosol particles in the bag. But we have given the molar equivalents for acrolein and N in the bag (0.3 mmol acrolein and 5 mmol total ammonia in the acrolein-ammonia bag; 0.3 mmol acrolein and 4 mmol total ammonia in each of the two acrolein-ammonium bags). The absolute amount of $(NH_4)_2SO_4$ or $NH_4Cl$ is about 264 mg or 212 mg. About 4 g of $H_2O$ (2 g from the ammonium aerosols and 2 g from the acrolein solution) was added into the bag. We have revised the experimental section accordingly.

(4) Sect 2.1: It would be helpful to state the total ammonia ($NH_3 + NH_4$) in each experiment.

We have used the term "total ammonia" to state the added amount of reduced nitrogen in the revised manuscript.

Revised Lines 18-19, Page 4: *We referred to the added nitrogen in this work as "total ammonia" (including both the ammonia $NH_3$ and ammonium ions $NH_4^+$).*

(5) Page 5 lines 5-6: What exactly is meant by "more than enough water"? How much liquid water is estimated to be in the chamber?

About 4 g of $H_2O$ was added into the ammonium system. Even if the RH reached 100%, the water vapor should be 17 $g/m^3$ (at 20°C), indicating no more than 1.7 g $H_2O$ as vapor in the 100L bag. Therefore, there were still near 2.3 g $H_2O$ will sustain a liquid film on the bag wall. That is what we mean by "more than enough water". We specified the actual numbers of added water in the experimental section.

(6) Page 5 lines 7-8: If the aerosol deposited to the wall, this would be observed in the APS data. Was that the case?

We did not use APS to measure the aerosol size in the acrolein-ammonium bag experiment. Instead, the APS data was obtained in a parallel set up. In this parallel experiment, the aerosol from atomizer with ammonium solution and about 100L $N_2$ was quickly filled into a clean Tedlar bag. After the filling, we used the APS to measure the size distribution of the ammonium aerosol in the bag. The filling of aerosol and $N_2$ costs several minutes and the APS measurement was done in about 15 minutes. We used this APS data to represent the size distribution of ammonium aerosol particles in the reaction. Though the deposition to the bag wall was inevitable even in this ~20 minutes, it was much shorter than the time the acrolein-ammonium bag was stored prior to the GC-MS and ESI MS analysis. We clarified this in the caption of Figure S1 in the supplement.

(7) Page 5 lines 9-11: "…in order to investigate the gas-phase reaction and liquid-phase reaction respectively." The measurement of a compound in the gas-phase does not necessarily imply that it was formed via a gas-phase mechanism. It could be formed in the condensed-phase and repartition.

We agree, and we revised the manuscript accordingly. We have deleted the phrase "in order to investigate the gas-phase reaction and liquid-phase reaction respectively" in the revised manuscript.

(8) Page 7 line 19: Do the authors have a suggestion for why pyran aldehyde was no longer observed?

We have carefully checked the GC-MS results for ammonium system. There is indeed a very small GC peak (overwhelmed by the neighbor picoline peak, unable to be auto-recognized by the GCMS analyzing system) corresponding to the dimer. We made it clearer in the revised manuscript. "*Pyran aldehyde was still observed, but as an inconspicuous peak, as shown in Figure 1(b).*" in revised Line 19, Page 7.

(9) Page 10 line 22: I would think that the neutral species rather than the cation would be more reactive.

Thank you for noticing this. The reviewer is correct. We modified the reaction 3 to show that it is the neutral form that is reacting, and changed the text on page 10 accordingly.

Revised Lines 18-21, Page 10: *We propose the following pathway to $(C_3H_4O)_2C_6H_8N^+$: Reaction*

*2 in Figure 2 leads to the formation of 3-methylpyridine, and then the lone pair on the pyridine nitrogen can attack the electrophilic site in the carbonyl of the acrolein dimer, to change the carbonyl to hemiaminal and form pyridinium compounds $(C_3H_4O)_2C_6H_8N^+$ (Reaction 3 in Figure 2).*

(10) Sect 3.3: This section should include a description of how one expects the chemistry to differ in bulk vs aerosol reactions (i.e., with regards to partitioning of compounds).

We added several sentences to discuss the reason for the difference between the bag-wall sample and bulk solution with similar pH value. In the revised manuscript, we pointed out that the possible more complete reaction of some volatile products in the bulk, which could not evaporate into the gas phase as sufficiently as they could in the bag experiments.

Revised Lines 15-20, Page 12: *This difference could be due to the more complete reaction in the bulk experiments. Since the bulk solution was sealed in a bottle with a relatively small head-space to undergo the aqueous-phase reactions, the volatile or semi-volatile products or intermediate products could not evaporate into the gas phase as sufficiently as they could in the Tedlar bag experiments. This enhanced the contribution of volatile and semi-volatile compounds to the reaction, resulting in the observed difference between the bag-wall residue sample and the bulk sample.*

(11) Page 11 line 6: I believe the Jang et al. (2003) reference looked at acid-catalyzed reactions of carbonyls. I don't think that reactions of the carbonyls with ammonia were considered.

In the original manuscript, we cited Jang's work, just to illustrate that most heterogeneous reactions are acid-catalyzed. Jang's work is about the reaction of carbonyls, but nothing with ammonia. Therefore, in the revised manuscript, we not only removed this citation, but also rewritten the sentence with more relevant citations.

Revised Lines 7-9, Page 11: *The formation of imidazole chromophores and other light-absorbing NOC in reactions involving ammonia has been demonstrated to be pH-dependent by several reports (Kampf et al., 2012; Phillips et al., 2017; Yu et al., 2011).*

(12) Page 12 lines 4-6: I don't necessarily see from the figure the decreasing signal intensity at 4.8 min. It appears that it decreases relative to the other peaks, but without a scale I don't know how it relates in terms of absolute signal.

The original sentence stated that the peak at 4.8 min was the predominant peak at pH 8, but not at pH 5-7. We did not mean to compare the signal intensity of 4.8 min peak at pH 8 with that at pH 5-7. In the revised manuscript, the ambiguous phrase "they were not as abundant as in the alkaline solutions, which can be revealed by the decreasing signal intensity at 4.8 min" was deleted.

(13) Page 12 line 10: Why not compare to pH 5?

The MS results for bulk solution of pH 6 and for that of pH 5 were very similar. We added "pH 5" in the revised sentence (Line 13, Page 12).

(14) Figure 5: It would be helpful to label the peaks (e.g., label the peak at 1.2 min as propylene imines).

We have labeled the major peaks in the revised Figure 5.

(15) Page 14 line 3-4: "…potential to form light-absorbing…" the pH of aerosols and droplets needs to be discussed before this conclusion can be made.

The formation of heterocyclic chromophores depends on the pH value. In the revised manuscript, we did not state the "light-absorbing" in the beginning of the Section 3.4. Products in aqueous phase with different pH were stated first, and then the case for the moderately acid aerosol was mentioned in the revised version.

Revised Lines 2-5, Page 15: ***Though the reported pH values of atmospheric aerosol particles vary from 0 to 9 (Guo et al., 2015; Hennigan et al., 2015; Pszenny et al., 2004; Weber et al., 2016), alkaline conditions are much less common. Even under the ammonia-rich conditions, such as those found in the northern China winter haze, the fine particles are moderately acidic with pH of around 5 (Song et al., 2018).***

(16) Page 14 line 12: Please see major comment 1.

The sentence has been removed in the revised manuscript.

(17) Page 15 line 16-17: This sentence is currently unsupported by the atmospheric implications section. It should be removed or the atmospheric implications section should be substantially expanded and made more quantitative.

The sentence has been revised and the improper phrases such as "the important sink" and "the important building blocks" have been deleted.

Revised Lines 22-24, Page 15: ***Since ammonium widely exists in the atmospheric aerosols, our work suggests that the reactive uptake of acrolein into aerosols is a potential atmospheric loss process for acrolein, which leads to the formation of nitrogen-containing, light-absorbing, heterocyclic SOA compounds.***

(18) Figure 9: I don't find this figure particularly helpful. The reaction pathways have already been discussed.

Figure 9 was used to summarize our study clearly, and we had ever revised this figure according to the suggestion by the co-editor before the manuscript was submitted as the discussion paper. So we continue to have this figure in the revised manuscript. Please the editor decides whether it need to be removed.

(19) General: There are numerous grammatical errors, many dealing with use of definite articles and subject-verb agreement. Correction of these errors would improve readability of the paper.

We have corrected as many grammatical errors and typos as we could. We hope the rest will be caught at the copy-editing stage.

(20) General: The resolution of the figures should be improved.

We offer the high-resolution (600dpi) tif. High resolution figures are shown in the manuscript.

---

## Author Response (AR2)

Dear Editor,

We have corrected the phrases in Line 24 of Page 8. "acrolein-ammonium" has been changed to "acrolein-ammonia", and "gaseous ammonium" has been changed to "ammonia". Both are marked in red in the manuscript.

We have renewed the Figure 9 replacing the "R" with a "H".

Many thanks for your help.

Xin YANG
Department of Environmental Science & Engineering